# Video-Mined Task Graphs for Keystep Recognition in Instructional Videos

**Kumar Ashutosh**
UT Austin and FAIR, Meta

**Santhosh Kumar Ramakrishnan**
UT Austin

**Triantafyllos Afouras**
FAIR, Meta

**Kristen Grauman**
UT Austin and FAIR, Meta

## Abstract

Procedural activity understanding requires perceiving human actions in terms of a broader task, where multiple keysteps are performed in sequence across a long video to reach a final goal state—such as the steps of a recipe or a DIY fix-it task. Prior work largely treats keystep recognition in isolation of this broader structure, or else rigidly confines keysteps to align with a predefined sequential script. We propose discovering a *task graph* automatically from how-to videos to represent probabilistically how people tend to execute keysteps, and then leverage this graph to regularize keystep recognition in novel videos. On multiple datasets of real-world instructional videos, we show the impact: more reliable zero-shot keystep localization and improved video representation learning, exceeding the state of the art. Project Page: https://vision.cs.utexas.edu/projects/task_graph/

## 1 Introduction

Instructional "how-to" videos online allow users to master new skills and everyday DIY tasks, from cooking to crafts to sports [51]. In the future, AR assistants able to parse such procedural activities could augment human skills by providing interactive guidance throughout the task in sync with the user's visual context [25, 56, 64], or by automatically creating video summaries of the most important information [2, 23, 54]. Similarly, human expert demonstration videos have the potential to steer robot behavior in the right direction for complex sequential tasks that have notoriously sparse rewards [52, 62, 81].

In a procedural activity, there is a single task goal that a person accomplishes by executing a series of *keysteps*, some of which have causal dependencies. For example, to make tiramisu, keysteps include *whisk the eggs*, *lay out the ladyfingers*, *sprinkle the cocoa*—and the cocoa must be sprinkled only after laying ladyfingers in the pan; to replace a bike tube, the wheel needs to be removed from the bicycle, then the old tube deflated and removed before the new one is fitted in. Thus, unlike mainstream video recognition tasks focused on naming an action in a short video clip [13, 14] like *shaking hands*, *playing instruments*, etc., procedural activity understanding requires perceiving actions in terms of the broader goal, breaking down a long-form video into its component keysteps.

To address keystep recognition, prior work has proposed creative ideas to either match video clips to keystep names based on the transcribed narrations [45, 55] or align visual steps to a rigid linear script (e.g., with dynamic time warping) [15, 18, 19, 36]. Other methods pose keystep recognition as a classification problem and label each fixed-sized chunk into one of the possible classes [48, 50, 77, 79, 80, 88]. However, these existing approaches face important limitations. Simple feature-keystep similarity matching and classification assume all actions are independent and can appear anywhere throughout the video, while the existing alignment models assume every keystep will match some

37th Conference on Neural Information Processing Systems (NeurIPS 2023).

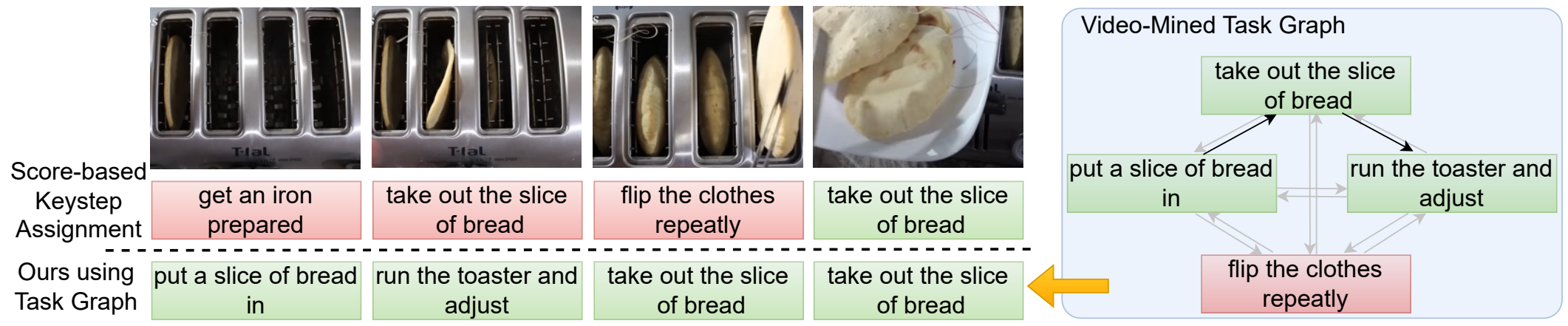

Figure 1: Our method uses a video-mined task graph as a prior to update the preliminary keystep assignments. The visual signals confuse a toaster with iron and bread with clothes, whereas our method focuses on the broader context using the task graph and assigns the correct keystep labels. When the visual signal predicts *taking out bread*, we instead correctly map it to *running the toaster* since taking out can only happen after putting the bread and running the toaster.

video segment—neither of which holds universally. Furthermore, the status quo is to rely on *text-based* knowledge bases to represent the likely order of keysteps [19, 45, 55], e.g., WikiHow, which fails to encapsulate the rich variety of ways in which the procedure may be executed in practice.

We propose to regularize keystep predictions on novel videos based on a *task graph* automatically mined from how-to videos. The task graph is a probabilistic data structure capturing the keysteps (nodes) and their dependency relationships (edges). First, we show how to automatically discover the graph structure directly from unlabeled, narrated how-to videos. Notably, video demonstrations are much richer than a scripted list of (text) steps like WikiHow; they show *how* to perform the task and, through different video instances, reveal the strength of dependencies between steps and the alternative ways in which the same goal can be reached. Next, we perform keystep recognition on novel videos using the task graph as a prior for task execution. Here we develop a beam-search approach in which keysteps confidently recognized from the visual and/or narration inputs are anchors, and subsequent less-confident steps are inferred leveraging the graph to find a high probability path. The task graph allows our model to "see the forest through the trees," since it can anticipate the overall arc of progress required to complete an instance of the full task.

Ours is the first work to use task graphs to enhance keystep prediction in instructional videos, and the first to discover a probabilistic task graph model directly from video. We first demonstrate that our novel approach improves zero-shot keystep localization for two challenging procedural activity datasets, COIN [67] and CrossTask [89], outperforming prior work [18, 19, 45, 79]. We also show that our graph learned from real-world videos surpasses the prior supplied by traditional written scripts. Next, we show that our task graph improves video representation learning for the large-scale HowTo100M dataset [51], where our corrected keystep pseudo-labels benefit pretraining compared to state-of-the-art keystep understanding work [45] and popular multimodal embeddings [50, 79]. In addition, we show the resulting pretrained video representations benefit multiple downstream tasks: keystep classification, keystep forecasting, and task classification. Finally, visualizing our learned task graphs, we show their ability to discover ties even across different tasks.

## 2 Related Work

**Keystep Recognition and Localization.** Keystep recognition for instructional video is a focus for multiple recent influential datasets [51, 67, 89]. COIN [67] and CrossTask [89] offer manually annotated keysteps for more than 900 keystep labels and 198 diverse procedural tasks. The even larger HowTo100M dataset [51] covers tens of thousands of tasks; while not labeled for keysteps due to its massive size, recent work shows the promise of (noisily) localizing keysteps in HowTo100M using a text-based matching between keysteps from a knowledge base (e.g., WikiHow) and the words spoken in the how-to video narration [45, 55, 86]. Our approach builds on this idea when mining videos to form a task graph.

Grounding keysteps in instructional videos [6, 19, 20, 21, 50, 51, 79, 89] is crucial for procedural planning [9, 12, 16, 40, 61, 71, 83, 85] and learning task structure [55, 87]. Some prior work localizes the keysteps by finding a similarity score between keystep (text) embeddings and video features using a multimodal embedding [50, 79], while others learn an embedding to map corresponding keysteps close together [6]. Unlike our approach, this assignment ignores the broader context about

the keysteps and their sequence. Another line of work uses Dynamic Time Warping (DTW) to ground keysteps to video [11, 15, 19, 40]. These methods take as input an ordered list of keysteps and localize them in the video while preserving the given ordering. While this has the potential advantage of enforcing a known ordering, existing methods do so rigidly: the single linear order is required, and it is assumed that all the actions are guaranteed to be present in the video—a very strong assumption for unconstrained video. While Drop-DTW [18] removes the last assumption by introducing a mechanism that allows dropping outliers, it remains constrained by the monotonic ordering. The set-supervised action recognition method [47] proposes a more relaxed pairwise consistency loss that uses weakly labeled videos and encourages the attentions on actions to follow a similar ordering. In contrast to any of the above, our proposed method uses a probabilistic task graph to guide the localization process and correct predictions based on the keystep transition patterns discovered in in-the-wild video demonstrations.

**Graphs for Video Understanding.** Prior work uses graphs to understand spatio-temporal relations between objects and actors in videos [7, 53, 74, 82]. Such modeling helps to surface underlying relationships that may not be captured implicitly by models. Graph Convolutional Networks (GCN) are another promising method to incorporate graph structure in the training process [30, 38, 53, 58]. Earlier work in activity recognition has explored a variety of statistical models to represent complex sequential activities, such as Bayesian Logic Networks [68], dynamic Bayesian Networks [60], AND-OR graphs [35], latent SVMs [39], and graph-parsing neural networks [57]. Our work is distinct for generating data-driven graphs from large-scale video samples, rather than exploit a predefined statistical model, and for bootstrapping noisy activity labels for recognition.

Methods for unsupervised procedure learning aim to discover the common structure from a set of videos showing the same task [2, 22, 59, 86] in order to reveal a common storyline [2, 22, 59] or in Paprika [86] (a concurrent approach) to generate pseudo-labels for video feature learning [86]. The goal of discovering the latent procedural structure resonates with our task graph formation; however, unlike the prior methods, we show how the task graph serves as an effective prior for accurate keystep recognition from video. Paprika [86] uses task graph nodes as a pretraining signal—their constructed graph is non-probabilistic, cannot represent that some transitions are more likely than others, and cannot be directly used for keystep recognition. Additionally, our approach predicts keysteps by fusing supervisory signals from the graph and a weakly-supervised similarity metric between video features and candidate keysteps, eliminating the need for explicit graph annotation [19, 64].

**Video Representation Learning.** Pretraining to extract meaningful visual representations is useful for many downstream tasks including action recognition [26, 33, 42, 43, 76] and action anticipation [1, 27, 28, 32, 49]. The pretraining objective is either to map the visual representations to meaningful classes [8, 45] or to its equivalent text representation [3, 44, 50, 79]. Text-guided pretraining further enables tasks like text-to-video retrieval [17, 24, 48, 79, 84] or video captioning [31, 48, 72, 78]. It is desirable to pretrain on large video datasets [13, 14, 34, 51] for better generalization. In particular, HowTo100M [51] is the largest instructional video dataset (134,472 hours of video); its scale makes annotations for keysteps impractical. Recent work uses external WikiHow (text) steps to generate keystep classes for pretraining [45, 55]. However, the generated keystep classes are noisy due to misalignments [37] and non-visual narrations [4], which affects the resulting representations. We show how to use a task graph to improve keystep label assignment across the massive dataset, which in turn significantly improves the resulting video representation and downstream benchmarks.

## 3 Technical Approach

We propose to discover task graphs from how-to videos and then use them as a prior for keystep recognition. Given a keystep vocabulary (in text only) and videos from various instructional tasks, our goal is to automatically localize any instances of those keysteps in each video.

To that effect, we first use external text corpora to obtain a keystep vocabulary. Next we obtain preliminary keystep labels by linking visual and/or narration representations between the video clips and candidate keystep names. Then we use those (noisy) keystep labels to construct a task graph, and finally we use the preliminary keystep labels and task graph prior to obtain high-quality keystep labels. The recognized and localized keysteps are themselves a useful output of the method. We also explore using the inferred keystep labels to support large-scale video representation learning for instructional videos. Fig. 2 contains an overview of the method.

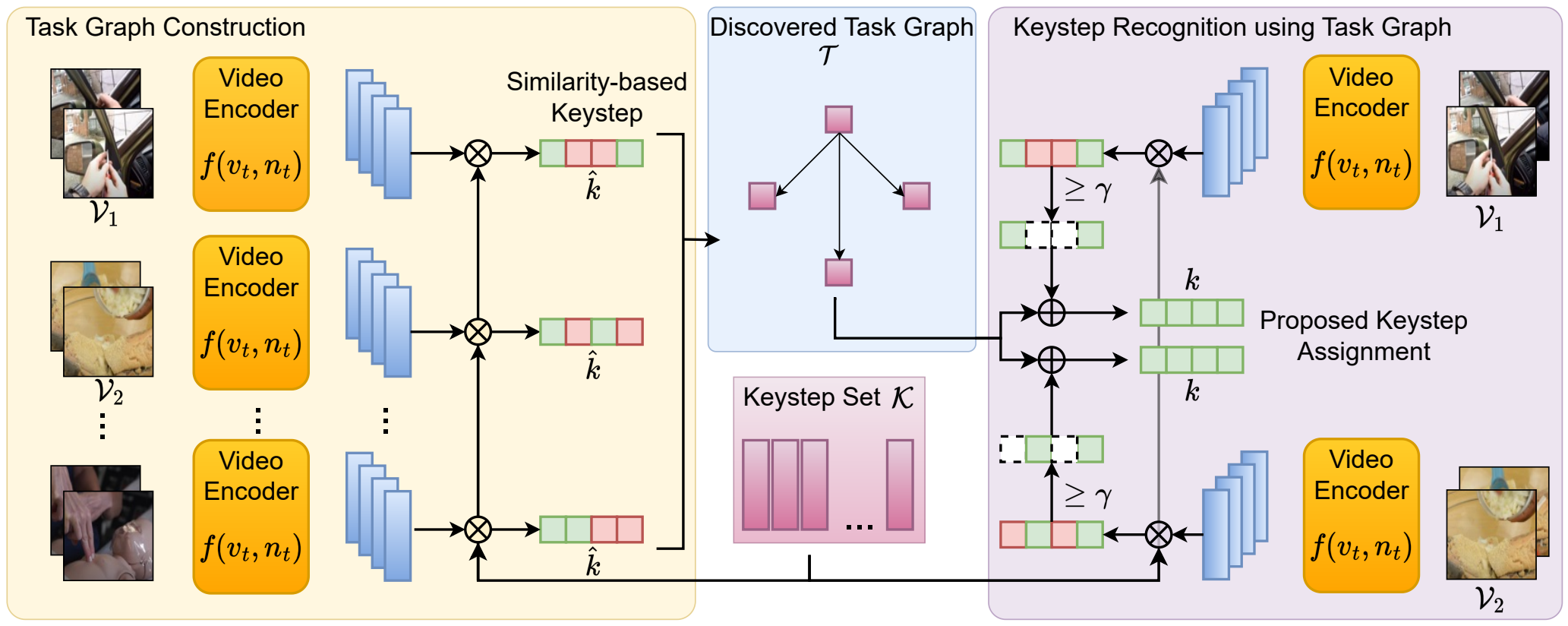

Figure 2: Overview of the proposed keystep recognition using our video-mined task graph. Left: We obtain similarity-based keystep $\hat{k}$ for each video clip in the dataset using similarity between keysteps in $\mathcal{K}$ and text and/or video features. Top middle: We use the inferred keystep labels to learn a probabilistic task graph $\mathcal{T}$. Right: Finally, we keep confident video segments and use the task graph $\mathcal{T}$ priors to obtain the final keystep predictions. See text.

**Task Definition.** We have two inputs: (1) an unannotated dataset of narrated instructional videos $\mathcal{D} = \{\mathcal{V}_i\}_{i=1}^{|\mathcal{D}|}$ and (2) a keystep vocabulary $\mathcal{K}$. Each video $\mathcal{V}_i$ in $\mathcal{D}$ is comprised of a sequence of clips, each of which has an associated spoken narration provided by the how-to demonstrator (and converted to text with ASR): $\mathcal{V}_i = \{(v_1, n_1), ..., (v_{|\mathcal{V}_i|}, n_{|\mathcal{V}_i|})\}$ where $v_t$ and $n_t$ are the visual and text segment corresponding to $t^{th}$ clip in the instructional video. The keystep vocabulary $\mathcal{K}$ is an unordered set of natural language terms for all the relevant keysteps in the tasks depicted in $\mathcal{D}$. For example, if the dataset contains the tasks "make tiramisu" and "fix bike derailleur", then $\mathcal{K}$ would contain terms like $\{$*whisk eggs*, *soak ladyfingers*, ..., *prop up the bike*, *shift bike to highest gear*,...$\}$. To give a sense of scale, datasets in our experiments contain 18 to 1,059 unique tasks and 105 to 10,588 unique keystep names. We do not assume the task labels are known per video. Note that words in the keystep vocabulary are not equivalent to the spoken narrations—the latter is unconstrained.

Our goal is to predict the correct keystep label $k_t \in \mathcal{K}$ for every $(v_t, n_t)$. For notation simplicity, we refer to localized keystep predictions as $k_t$ and keystep names in the vocabulary as $k_i$, for $i \neq t$.

**Sourcing a Keystep Vocabulary.** To obtain a keystep vocabulary $\mathcal{K}$, we consult existing text knowledge bases, namely WikiHow and the vocabulary curated by experts for the COIN dataset [67]. WikiHow [75] contains written instructions of more than 240K how-to tasks. Each article has a list of steps that needs to be taken to achieve the desired task. For example, *"Make Instant Pudding"*[1] has steps *"Tear open the instant pudding mix and pour it into the bowl"*, *"Pour the mixture into small serving bowls"*. Note that several tasks share common keysteps, e.g. *"Pour the mixture into small serving bowls"* can happen in many recipes. In our experiments, we either use keysteps provided in COIN [67] ($|\mathcal{K}| = 749$) and CrossTask [89] ($|\mathcal{K}| = 105$), or WikiHow keysteps for the HowTo100M [51] dataset ($|\mathcal{K}| = 10,588$). See [45] for details on keystep set curation and Supp. for details.

**Preliminary Keystep Assignment.** First we make a preliminary estimate of each $k_t$. Following recent work [45, 79], we draw on the similarity between (1) language features for the keystep names $k_i$ and (2) visual and/or language features for the video clip $(v_t, n_t)$. Specifically, let $f_v := \mathbb{R}^{H\times W\times C\times T} \rightarrow \mathbb{R}^D$ and $f_n := \mathbb{R}^L \rightarrow \mathbb{R}^D$ be the $D-$dimensional visual encoder and text encoder, respectively. Here, $(H, W, C, T)$ denotes (height, width, channels, time) and $L$ is the maximum language token length. Correspondingly, for every segment $(v_t, n_t)$ we obtain feature vectors $f_v(v_t)$ and $f_n(n_t)$. Since keystep names are themselves text sentences, we also use $f_n(k_i)$ to obtain each keystep embedding. Here we denote $f(v_t, n_t)$ as a generic feature extractor that can be either $f_v$ or $f_n$ or a combination of both (to be specified per experiment). We obtain the preliminary keystep assignments as $\hat{k}_t = \text{argmax}_{k_i \in \mathcal{K}} f(v_t, n_t)^T f_n(k_i)$. Next, we will use these (noisy) keystep estimates to construct our task graph, which in turn will allow us to improve our keystep assignments

[1] https://www.wikihow.com/Make-Instant-Pudding

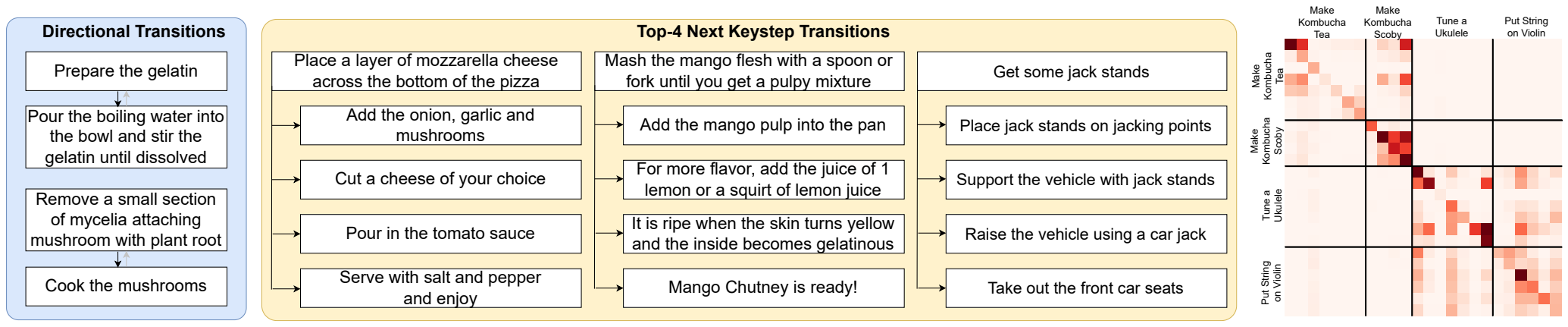

Figure 3: Visualization of properties emerging from the video-mined task graph. Some of the keysteps are directional e.g., *pouring gelatin into boiling water* must be preceded by the keystep *prepare gelatin* (left). Some examples of top-4 next keysteps (center), and a visualization showing keystep sharing discovered across related HowTo100M tasks (right). Best viewed in zoom.

for all clips. These initial estimates $\hat{k}_t$ will also serve as a baseline, corresponding to what is done in practice today by existing multimodal methods using either visual [79] or narration text [45] features.

**Video-Mined Task Graphs.** Having obtained a rough keystep assignment, we next construct a task graph to capture how keysteps are related to each other. Our task graph design is motivated by two key observations. First, keysteps can share different relationships. Some keystep pairs are *directional*, that is, one is the precursor to the other (e.g., *whisking an egg* must be preceded by *breaking an egg*); others are simply *connected*, meaning they often both appear in an activity but with less constrained temporal ordering (e.g., *whisking egg* and *sifting flour*), while others are *unconnected*, meaning they support different task goals altogether (e.g., *mixing the salad* and *cutting wood with a saw*). Second, multiple video demonstrations of the same task share the same high-level objective and hence similar keysteps, yet not necessarily in the same temporal order. For example, two how-to's for making pizza may use different ingredients and perform steps in a different order.

Hence, rather than hand-design the task graph [19, 64], we propose to mine it directly from video samples. In addition, rather than encode possible transitions uniformly [86], we propose to represent them probabilistically. Finally, rather than constrain the graph to be representative of a single task [19], we aim to discover a single task graph across all tasks, such that related tasks can share information (e.g., *whisk eggs* would be common to many different recipes, and any sequential patterns surrounding it could inform other tasks in the graph). See Fig. 3 (right).

Formally, the task graph $\mathcal{T} = (V, E, w)$ has all the keysteps as nodes, i.e., $V = \mathcal{K}$ and all transitions as edges, $E = \mathcal{K} \times \mathcal{K}$. Naturally, we want the graph to be directed and weighted so that more probable keystep transitions have higher weights. For any pair of keysteps $k_i$ and $k_j$, we define the weights of the edges as:

$$w(k_i, k_j; \mathcal{T}) = \frac{\sum\limits_{\mathcal{V} \in \mathcal{D}} \sum\limits_{t \in |\mathcal{V}|} \mathbb{1}(\hat{k}_t = k_i, \hat{k}_{t+1} = k_j)}{\sum\limits_{\mathcal{V} \in \mathcal{D}} \sum\limits_{t \in |\mathcal{V}|} \sum\limits_{x \in |\mathcal{K}|} \mathbb{1}(\hat{k}_t = k_i, \hat{k}_{t+1} = k_x)}$$

where $\mathbb{1}$ is the indicator function. In other words, the weight of an edge between $k_i$ and $k_j$ is the count of transitions between the two keysteps normalized by total count of keysteps $k_i$ being executed. The normalization converts the weight into probability distribution and the sum of all outgoing edges is 1, i.e., $\forall k_i \in \mathcal{K}, \sum_{k_x \in \mathcal{K}} w(k_i, k_x) = 1$.

Fig 3 (left) shows interesting properties emerging from the task graph. Some pairs of keysteps are *directional*, e.g., only after *preparing the gelatin* can it be poured into the *boiling water*. Fig 3 (middle) shows example top-4 transitions—all of which show the expected trend. Finally, the keystep transition heatmap (right) shows that the task graph discovers transitions between keysteps of related tasks like (*tuning ukulele*, *fixing violin string*) or (*kombucha tea*, *kombucha scoby*), while also detecting that neither pair shares with the other. We stress that these inter-task relationships are a direct consequence of our video-mined, probabilistic keystep transitions, unlike [19, 64, 86].

**Keystep Update Using the Video-Mined Task Graph.** Next, we use the preliminary keystep labels $\hat{k}_t$ and task graph $\mathcal{T}$ to obtain corrected keystep labels $k_t$. Intuitively, we want the task graph to regularize the initial estimates: keep keystep labels with a strong signal of support from the visual and/or text signal in the video, but adapt those keystep labels with low confidence using the prior given by the graph structure $\mathcal{T}$.

Specifically, we estimate the confidence score for keystep $\hat{k}_t$ as the similarity between the video feature(s) $f(v_t, n_t)$ and the keystep name feature $f_n(k_i)$: $s(\hat{k}_t) = f(v_t, n_t)^T f_n(\hat{k}_t)$. Given any high-confidence pair $\hat{k}_{t^-}$ and $\hat{k}_{t^+}$, where $t^-$ and $t^+$ are the closest high-confidence time instances before and after $t$, respectively, we find the highest probability path in $\mathcal{T}$ between $\hat{k}_{t^-}$ and $\hat{k}_{t^+}$. Formally,

$$k_t = \begin{cases} \hat{k}_t & \text{if } f(v_t, n_t)^T f_n(\hat{k}_t) \geq \gamma \\ \text{PathSearch}(\hat{k}_{t^-}, \hat{k}_{t^+}, t) & \text{otherwise} \end{cases}$$

where $\gamma$ is a confidence threshold, and $t^-$ and $t^+$ are time instances with confidence more than $\gamma$. PathSearch finds the maximum probability path between $\hat{k}_{t^-}$ and $\hat{k}_{t^+}$ from the task graph $\mathcal{T}$. We convert all the probabilities to their negative log and use Dijkstra's algorithm to find the minimum weight path. To account for self-loops, we prepend and append $\hat{k}_{t^-}$ and $\hat{k}_{t^+}$, respectively to the resulting path. Since the obtained path can have a variable number of keysteps, we assign the discovered keysteps uniformly between $t^-$ and $t^+$ (see Supp. for an example).

The idea of PathSearch is inspired from Beam Search algorithms, commonly used in Machine Translation [5, 10, 29, 65] for pursuing multiple possible sequential hypotheses in parallel, where we assume $k_{t^-}$ and $k_{t^+}$ as the start and end token. Instead of finding the maximum probable token (keystep in our context), Beam Search finds a sequence of tokens that maximize the overall likelihood of the selected token sequence. The above process yields corrected keysteps $k_t$ that we use as our predicted keystep label.[2] Fig. 4 shows some qualitative results. Note that PathSearch algorithm's time complexity is $O(|\mathcal{K}|^2)$ that introduces a minimal overhead. In general, this overhead is even less than a forward pass to the model.

**Keystep Localization and Instructional Video Representation Learning.** We explore the effectiveness of our approach in two settings: zero-shot keystep localization and representation learning for instructional videos. For zero-shot keystep localization, we evaluate the accuracy of our model's final keystep predictions compared to withheld ground truth labels. We stress that the results are zero-shot, since we are provided no clips annotated with their keysteps.

For representation learning, we augment the original unannotated dataset $\mathcal{D}$ with pseudo-labeled keysteps to create $\mathcal{D}' = \{\mathcal{V}_i\}_{i=1}^{|\mathcal{D}|}$, where $\mathcal{V}_i = \{(v_1, n_1, k_1), ..., (v_{|\mathcal{V}_i|}, n_{|\mathcal{V}_i|}, k_{|\mathcal{V}_i|})\}$ are the clips, narrations, and our model's inferred pseudo-labels. Our objective is to learn a video representation $F_V(v; \theta) := \mathbb{R}^{H \times W \times C \times T} \rightarrow \mathbb{R}^{|\mathcal{K}|}$ such that $\text{argmax}_{i \in |\mathcal{K}|} F(v_t; \theta) = k_t$. Here $(H, W, C, T)$ denote (height, width, channels, time) in a video segment. This is a standard classification problem and we choose cross entropy as the training loss.

To evaluate the quality of the resulting learned representation $F_V$, we consider several downstream tasks. First, we evaluate *task classification* on $\mathcal{D}$ against the withheld ground truth task labels (e.g., is the video depicting "make tiramisu" or "make omelet"), a common task for feature learning in instructional video [3, 8, 45] that can be evaluated more readily on large-scale datasets given the availability of task labels (vs. unavailability of keystep labels). In addition, we evaluate *keystep classification* given the temporal region of a keystep and *keystep forecasting*, where we must anticipate the next keystep given the video observed so far.

**Network Architecture and Implementation Details.** For zero-shot keystep recognition $f_v$ and $f_n$ are 768-dimensional frozen visual and text encoders of VideoCLIP [79]. For representation learning, we use MP-Net [63] to compute sentence embeddings and a modified TimeSformer [8] with $10,588$ output classes as the visual encoder. The weights of the visual encoder are initialized to Kinetics [14] pretraining. For downstream tasks, we freeze the backbone and only replace the last linear layer with the dimension of the classification task, namely 778 and 133 output classes for keystep classification and forecasting in COIN and CrossTask, respectively, and 180 and 18 classes for task classification.

For zero-shot keystep segmentation, we use $\gamma = 0.5$ and $\gamma = 0.3$ for text and video features, respectively, since video features offer stronger supervision. We observe that the performance is not sensitive for $\gamma \in [0.3, 0.5]$; details in Supp. In experiments where we consider both the video and text modalities, during path search, in the case of conflicting keystep suggestions, we choose the video

[2]We also explored a variant using Bayes Recursive Filtering (BRF) [69] to update the keystep belief state, but found this model less effective, likely because BRF only looks backward in time thus using less temporal context. See Supp. for experimental comparisons.

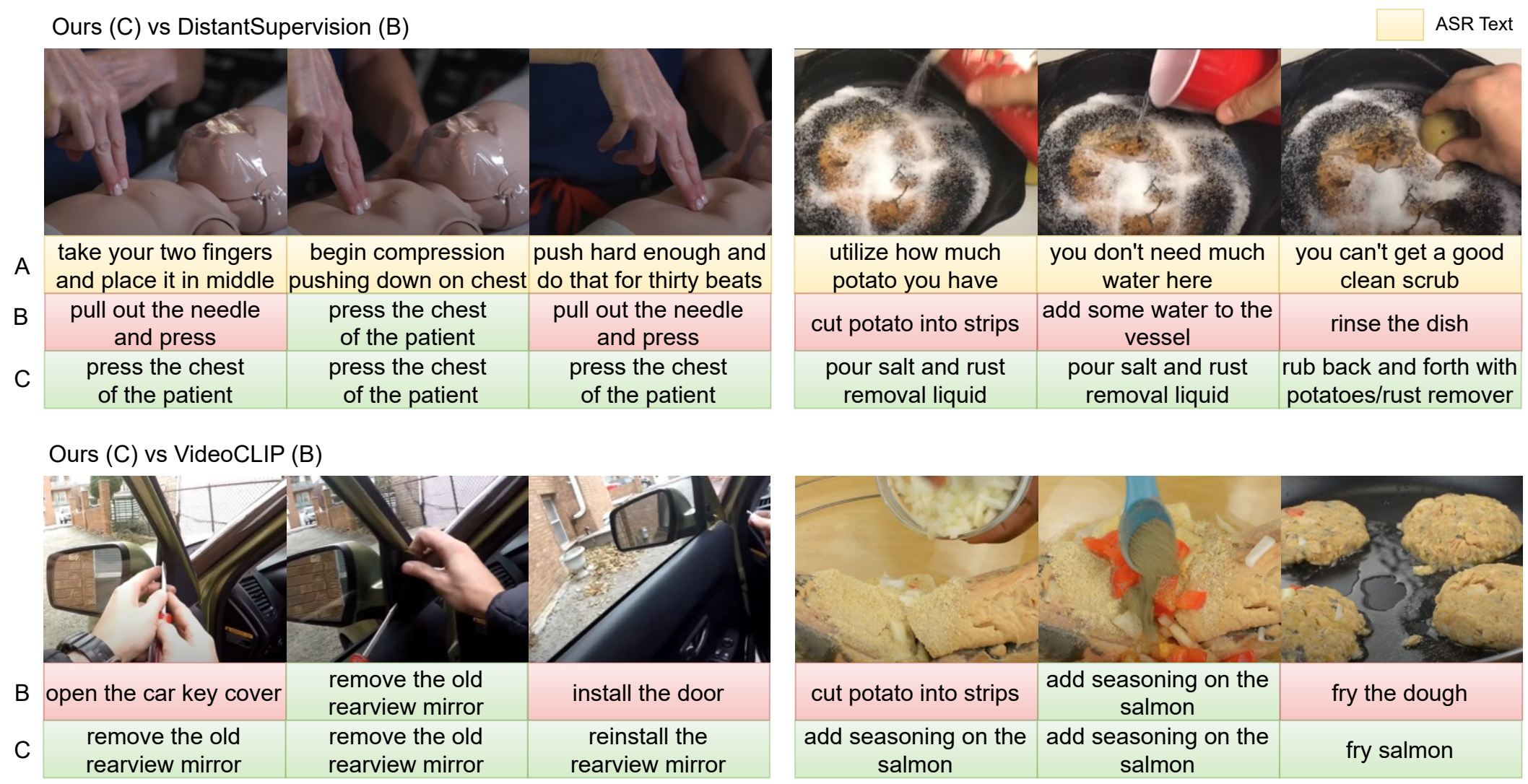

Figure 4: Qualitative examples comparing our keystep recognition with DistantSupervision [45] (first row) and VideoCLIP [79] (second row). Red/green denote incorrect/correct keystep predictions; yellow shows ASR narrations. Our proposed method is effective in correcting preliminary keystep assignments and reasons about the overall task structure. Best viewed in color and zoom.

keystep since the visuals tend to be a stronger cue. See Supp. for experiments justifying this design. For representation learning, similar to [45], we train the video model for 15 epochs with SGD with learning rate $5 \times 10^{-3}$ followed by 15 epochs with AdamW [46] with learning rate $5 \times 10^{-5}$. In both cases, the learning rate is decayed progressively by 10 times in epochs 11 and 14.

## 4 Experiments and Results

**Datasets.** We use three public datasets of instructional videos—COIN, CrossTask, and HowTo100M—all of which were compiled from in-the-wild data on YouTube, and are accompanied by ASR transcriptions of the YouTuber's spoken narrations ($n_t$). COIN [67] and CrossTask [89] contain 11,827 and 2,750 instructional videos, respectively, and are annotated for 778 and 133 (749 and 105 *unique* keysteps, respectively) keystep labels spanning 180 and 18 tasks, respectively. We use clip labels for evaluation only. HowTo100M [51] contains 1.22M instructional videos (i.e. more than 100$\times$ larger than the others) spread over 23,000 tasks. As noted above, we leverage WikiHow [75] to obtain the keystep vocabulary $\mathcal{K}$ for HowTo100M. We use COIN and CrossTask for both zero-shot keystep recognition and downstream task evaluation of our pretrained features. We use HowTo100M for downstream tasks only, since it lacks ground truth keystep annotations.

**Evaluation Metrics**. For keystep segmentation, we evaluate Frame-wise Accuracy (Acc) and Intersection over Union (IoU), following [18, 19, 79]. For every keystep $k_i$, Frame-wise Accuracy is the fraction of frames with ground truth $k_i$ that has the correct assignment. The overall Frame-wise Accuracy is the mean accuracy for every keystep. Likewise, IoU is the intersection over union of a keystep $k_i$, averaged for every $i$. Consistent with prior work [18, 19], we do not consider background frames. For keystep classification, task classification, and keystep forecasting, we use the standard accuracy metric.

### 4.1 Zero-Shot Keystep Recognition

In this task, we predict keystep $k_t$ for each time instance $t$ in a given video $\mathcal{V}$ in the test set. Following [79], we make one keystep prediction per second. We compare the following competitive baselines:

**DistantSupervision [45].** A state-of-the-art model for zero-shot keystep recognition that predicts keysteps based on the text feature similarity between narrations and keysteps, i.e., $f(v_t, n_t) = f_n(n_t)$.

Table 1: Zero-shot keystep recognition on COIN and CrossTask for three modality choices—text, video and video-text. We outperform strong baselines on all tasks. '-' means the method is n/a.

| | Text-only | | | | Video-only | | | | Video-Text | | | |
|---|---|---|---|---|---|---|---|---|---|---|---|---|
| | COIN | | CrossTask | | COIN | | CrossTask | | COIN | | CrossTask | |
| Method | Acc | IoU | Acc | IoU | Acc | IoU | Acc | IoU | Acc | IoU | Acc | IoU |
| Random | 0.0 | 0.0 | 0.01 | 0.01 | 0.0 | 0.0 | 0.01 | 0.01 | 0.0 | 0.0 | 0.01 | 0.01 |
| VideoCLIP [79] | - | - | - | - | 13.2 | 4.0 | 28.5 | 6.5 | 13.3 | 4.0 | 28.5 | 6.5 |
| DistantSup. [45] | 9.8 | 3.0 | 16.1 | 3.7 | - | - | - | - | - | - | - | - |
| Linear Steps | 10.4 | 3.1 | 16.3 | 3.8 | 13.4 | 4.3 | 28.5 | 6.5 | 13.4 | 4.0 | 28.5 | 6.5 |
| Auto-Reg [66] | 10.2 | 3.1 | 16.4 | 3.7 | 13.6 | 4.3 | 28.5 | 6.5 | 13.7 | 4.2 | 28.5 | 6.5 |
| Pruning Keysteps | 11.3 | 3.4 | 16.4 | 3.8 | 13.4 | 4.2 | 28.5 | 6.5 | 13.5 | 4.1 | 28.5 | 6.5 |
| Ours | **16.3** ± 0.3 | **5.4** ± 0.1 | **20.0** ± 0.2 | **4.9** ± 0.1 | **15.4** ± 0.1 | **4.7** ± 0.1 | **28.6** ± 0.0 | **6.6** ± 0.0 | **16.9** ± 0.1 | **5.4** ± 0.1 | **28.9** ± 0.1 | **6.7** ± 0.0 |

Table 2: Task-level keystep recognition on CrossTask. We outperform all methods in accuracy despite lacking their privileged information.

| Method | Acc | IoU |
|---|---|---|
| Random | 11.0 | 4.7 |
| Bag of Steps | 20.5 | 13.7 |
| Drop-DTW [18] | 22.3 | 15.1 |
| Graph2Vid [19] | 24.8 | **16.8** |
| Ours | **30.5** ± 0.2 | 16.0 ± 0.3 |

Table 3: Task classification on HowTo100M. On both modalities we outperform all state-of-the-art methods.

| Method | Modality | Acc@1 | Acc@5 |
|---|---|---|---|
| MIL-NCE [50] | - | 6.2 | 19.7 |
| VideoCLIP [79] | - | 7.9 | 25.5 |
| DistantSup. [45] | Video | 9.5 | 30.1 |
| DistantSup. [45] | Text | 12.3 | 28.5 |
| Ours | Video | 14.0 ± 0.3 | 33.5 ± 0.2 |
| Ours | Text | **15.5** ± 0.2 | **35.0** ± 0.2 |

**VideoCLIP [79].** A popular transformer-based contrastive pretraining approach for zero-shot video-text understanding. To apply it here, we use the cross-modal video feature similarity with the keysteps, i.e., $f(v_t, n_t) = f_v(v_t)$.

**Bag of Steps.** This baseline (devised in [19]) is similar to the first two, except here the keystep set only contains known steps in a given task. Naturally, the number of candidate keysteps in this case is much lower than the keystep set used above and by our model.

**Auto-Regressive [66, 70].** Instead of constructing an explicit task graph, an implicit representation could also model dependencies between keysteps. We use a transformer network to revise the preliminary noisy keystep labels (previous baselines) based on their aggregation over time.

**Pruning Keysteps.** The keystep set $\mathcal{K}$ is generally broad and contains unlikely keysteps for some tasks, e.g., *unscrew the bolt* is irrelevant in the cooking task. We first cluster keysteps into $C$ semantically similar clusters using k-means on their keystep embeddings and assign each video to one of the clusters per average similarity of the video's clips with the cluster members. Then, we compute similarity between only the selected cluster's keysteps and the video features to infer the per-clip keystep labels.

**Linear Steps.** Instead of using task graph, this baseline uses a linear order of keysteps as given in the dataset annotations. We still use keystep set $\mathcal{K}$ for preliminary assignment.

**Drop-DTW [18].** A SotA DTW-based approach where a linear order of steps is assumed. It requires a known order for execution for each task in the dataset.

**Graph2Vid [19].** A SotA DTW-based approach that parses a non-probabilistic graph for each task and then performs DTW-based matching on all possible paths and chooses the one with the highest matching score.

Note that the Bag of Steps, Drop-DTW, and Graph2Vid models all have privileged information during inference compared to our model, namely the known task and its (ordered) list of keysteps. Hence below we compare those models in a separate experiment called *task-level keystep recognition* [19], where candidate keysteps for all methods come only from the subset of keysteps $\mathcal{K}_T$ per task $T$ as mined from WikiHow ($|\mathcal{K}_T| \ll |\mathcal{K}|$). In all other zero-shot experiments, we use the universal keystep vocabulary $\mathcal{K}$.

Table 4: Downstream evaluation on CrossTask and COIN for keystep recognition (SR), task recognition (TR), and keystep forecasting (SF), using either an MLP (left) or transformer (right).

| | MLP | | | | | | Transformer | | | | | |
|---|---|---|---|---|---|---|---|---|---|---|---|---|
| | CrossTask | | | COIN | | | CrossTask | | | COIN | | |
| Method | SR | TR | SF | SR | TR | SF | SR | TR | SF | SR | TR | SF |
| TSN [73] | - | - | - | 36.5 | - | - | - | - | - | - | 73.4 | - |
| ClipBERT [41] | - | - | - | 30.8 | - | - | - | - | - | - | 65.4 | - |
| S3D [50] | 39.9 | 87.4 | 20.1 | 37.5 | 68.5 | 19.9 | 45.3 | 87.8 | 21.7 | 37.3 | 70.2 | 28.1 |
| SlowFast [26] | 44.9 | 89.7 | 23.4 | 32.9 | 72.4 | 23.0 | 48.5 | 89.8 | 24.0 | 39.6 | 71.6 | 25.6 |
| VideoCLIP [79] | 51.3 | 94.7 | 24.2 | 39.4 | 82.9 | 30.0 | 60.1 | 92.3 | 26.0 | 51.2 | 72.5 | 34.6 |
| TimeSformer [8] | 55.5 | 95.0 | 25.9 | 48.3 | 87.0 | 32.7 | 60.9 | 93.8 | 27.1 | 54.6 | 88.9 | 38.2 |
| DistantSup. [45] | 58.4 | 96.1 | 28.3 | 54.1 | 88.2 | 35.5 | 64.2 | 95.2 | 29.7 | 57.0 | 90.0 | 39.4 |
| Ours | **59.5** $\pm$ 0.1 | **97.1** $\pm$ 0.1 | **29.5** $\pm$ 0.1 | **55.2** $\pm$ 0.1 | **89.4** $\pm$ 0.0 | **36.3** $\pm$ 0.1 | **64.5** $\pm$ 0.0 | **96.0** $\pm$ 0.1 | **30.2** $\pm$ 0.1 | **57.2** $\pm$ 0.0 | **90.5** $\pm$ 0.0 | **40.2** $\pm$ 0.1 |

**Results.** Table 1 shows the zero-shot results on COIN and CrossTask. Throughout, standard errors denote variation across different test data splits. In all the three variants (text-only, video-only, and video-text), our method outperforms strong baselines and prior work. We see a gain of up to 6.5% (relative 66%). Importantly, our model's preliminary keystep assignments correspond to the VideoCLIP [79] and DistantSupervision [45] baselines for the video-only and text-only settings; our relative gains directly show the impact of our video-mined task graph in correcting those labels. In addition, our gains over Linear Steps show the impact of our full probabilistic graph structure, compared to a prior based on linear ordering of steps.

Fig. 4 shows some qualitative examples comparing our outputs with DistantSupervision [45] (top) and VideoCLIP [79] (bottom). The narration *pushing for thirty beats* is incorrectly mapped to *using needle* by DistantSupervision, whereas VideoCLIP confuses *salmon* with *cloth*. Our task graph prior improves recognition.

Table 2 shows the task-level keystep recognition results. We outperform all methods, including the state-of-the-art Graph2Vid [19] by 5.7% (relative 23%) in accuracy. Our IoU is similar to Graph2Vid's. We observe that the baselines conservatively assign the background label whenever the confidence is low, which helps IoU since background is not counted in the union operation [18, 19]. In contrast, accuracy accounts equally for false negatives and false positives, making it a more complete metric. We also emphasize that unlike Drop-DTW [18] and Graph2Vid [19], we do not assume keystep ordering from WikiHow recipes since that restricts the applicability to general in-the-wild instructional videos. Overall, our method provides a significant advantage over all the strong baselines and achieves state-of-the-art keystep recognition.

### 4.2 Instructional Video Representation Learning for Downstream Tasks

Next we apply our model to learn features $F_V(v;\theta)$ on the large-scale HowTo100M and assess their impact for multiple downstream tasks. We add comparisons to other popular pretrained representations, MIL-NCE [50], VideoCLIP [79], and TSN [73].

Table 3 shows the task classification results compared to multiple state-of-the-art methods on the validation split of HowTo100M, the dataset we use for pretraining $F_V$. Following [45], we infer task labels by predicting keysteps then mapping them back to the task category. Our approach outperforms all the existing methods. Again, our gain versus VideoCLIP and DistantSupervision shows the impact of our improved pseudo-labels.

Table 4 shows the results when we transfer the features pretrained on HowTo100M to downstream tasks on COIN and CrossTask. We deploy both MLPs and transformers fine-tuned for each task.[3]

[3]Note that the concurrent work Paprika [86] uses a setting different from the SotA DistantSupervision [45], which makes their numbers not comparable; DistantSupervision's reported results are higher than those re-implemented in [86] for all tasks, making it a stronger baseline to beat. Further, keeping our setting consistent with DistantSupervision allows us to compare with other SotA methods.

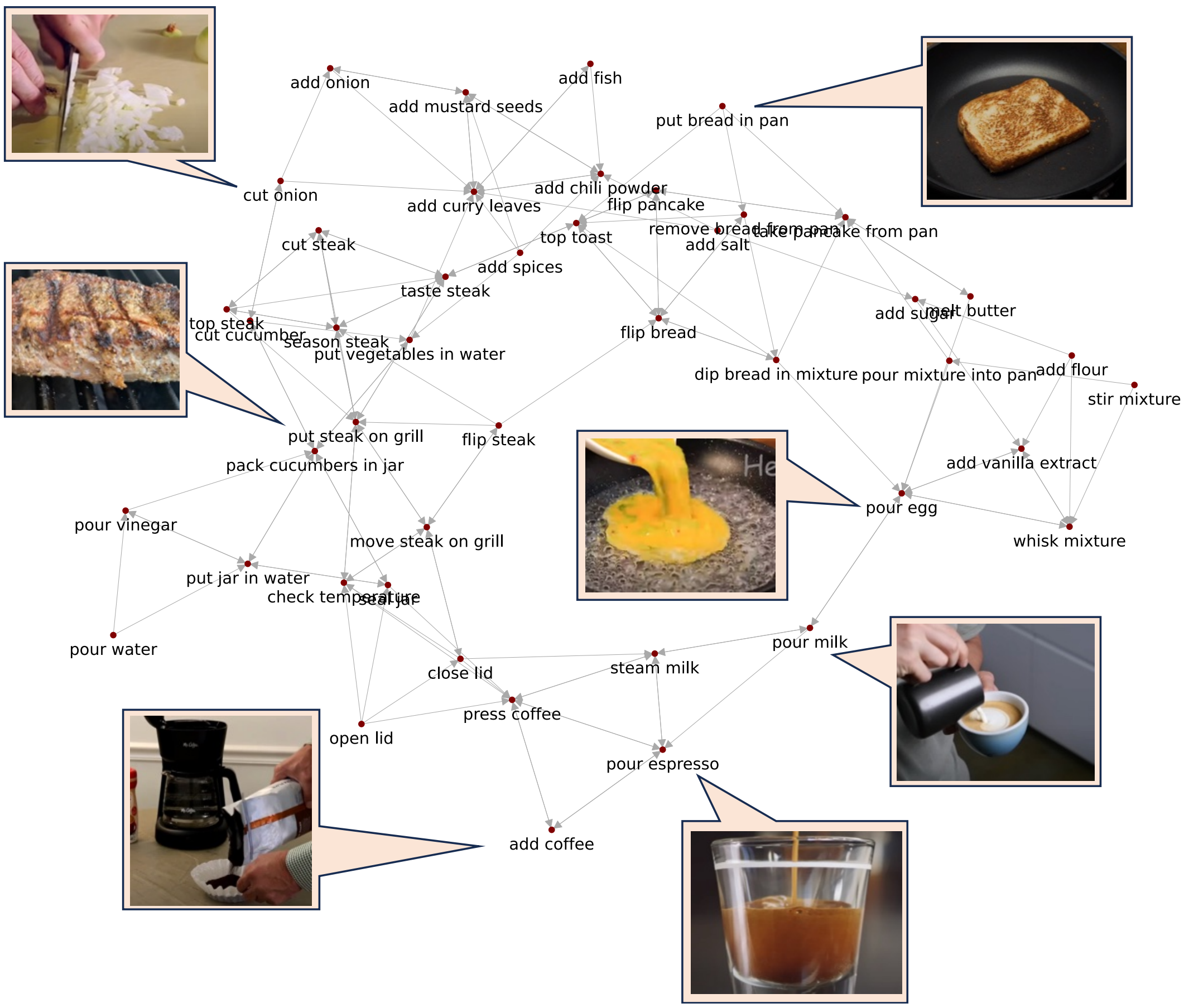

Figure 5: Task graph visualization of CrossTask (35/108 keysteps). We only show top 5 transitions (excluding self-loops) within this keystep subset for clear visualization. The edges are directed. Please zoom for best view.

Our results are strongest overall. We outperform the state-of-the-art DistantSupervision [45] on all downstream tasks for both datasets under both architectures.

### 4.3 Task Graph Visualization

We display a portion of the mined task graph for CrossTask in Figure 5. It contains 35 out of 108 keysteps with top 5 transitions labeled (to avoid clutter). We see some interesting properties. For example, "press coffee" happens only after "add coffee", "close lid" after "open lid" whereas "cut cucumber" and "cut onion" can both happen after each other. Again, this structure is discovered automatically from unannotated videos.

## 5 Conclusion

We introduced an approach to discover the structure of procedural tasks directly from video, and then leverage the resulting task graph to bolster keystep recognition and representation learning. Our model offers substantial gains over SotA methods, and our qualitative results also validate our improvements. In future work we plan to explore video-mined task graphs for other video understanding tasks including procedure planning and mistake detection.

## 6 Acknowledgement

UT Austin is supported in part by the IFML NSF AI Institute. KG is paid as a research scientist at Meta. We thank the authors of DistantSup. [45] and Paprika [86] for releasing their codebases.

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
