# *Supplementary Material for* Video-Mined Task Graphs for Keystep Recognition in Instructional Videos

**Kumar Ashutosh**
UT Austin and FAIR, Meta

**Santhosh Kumar Ramakrishnan**
UT Austin

**Triantafyllos Afouras**
FAIR, Meta

**Kristen Grauman**
UT Austin and FAIR, Meta

## 1   List of Contents

## 2   Sourcing a Keystep Vocabulary for Large-scale HowTo100M

HowTo100M is a large-scale dataset with more than 1M videos and 100M clips. Keystep annotation at this scale is infeasible. To obtain a feasible set of keysteps $\mathcal{K}$, DistantSupervision [1] collects step headlines from WikiHow corresponding to the video category having at least 100 videos in HowTo100M. This yields a set of 1059 video categories and 10588 keysteps collected from WikiHow articles corresponding to these task labels. We use the same set of keysteps in our downstream experiments.

## 3   Keystep Update Using the Video-Mined Task Graph: An Example

We use PathSearch algorithm to find the keysteps between the chosen anchor keysteps at time instants $t^-$ and $t^+$. As an example, if $t^- = 1$ and $t^+ = 10$ and the PathSearch outputs are $k_{d_1}, k_{d_1}, k_{d_3}$, we uniformly assign $k_t = \hat{k}_{t^-}$ for $1 \le t \le 2$, $k_t = \hat{k}_{d_1}$ for $3 \le t \le 4$, $k_t = \hat{k}_{d_2}$ for $5 \le t \le 6$, $k_t = \hat{k}_{d_3}$ for $7 \le t \le 8$ and finally, $k_t = \hat{k}_{t^+}$ for $9 \le t \le 10$. See the attached video for a visual example.

## 4   Sensitivity on Confidence Threshold for PathSearch

Our proposed algorithm uses a confidence threshold $\gamma$ for PathSearch anchors. We use $\gamma = 0.5$ and $\gamma = 0.3$ for text and video features, respectively (L243). We show that the performance of our method is not sensitive to the choice of $\gamma$ and choosing $\gamma$ anywhere in the range $[0.3, 0.5]$ yields a performance better than the strong baselines. Table 1 shows the performance of our method for

37th Conference on Neural Information Processing Systems (NeurIPS 2023).

Table 1: Zero-shot keystep recognition on COIN and CrossTask for three modality choices—text, video and video-text, **along with ablations**. We outperform strong baselines on all tasks. '-' means the method is n/a.

| | Text-only | | | | Video-only | | | | Video-Text | | | |
|---|---|---|---|---|---|---|---|---|---|---|---|---|
| | COIN | | CrossTask | | COIN | | CrossTask | | COIN | | CrossTask | |
| Method | Acc | IoU | Acc | IoU | Acc | IoU | Acc | IoU | Acc | IoU | Acc | IoU |
| Random | 0.0 | 0.0 | 0.01 | 0.01 | 0.0 | 0.0 | 0.01 | 0.01 | 0.0 | 0.0 | 0.01 | 0.01 |
| VideoCLIP [4] | - | - | - | - | 13.2 | 4.0 | 28.5 | 6.5 | 13.3 | 4.0 | 28.5 | 6.5 |
| DistantSup. [1] | 9.8 | 3.0 | 16.1 | 3.7 | - | - | - | - | - | - | - | - |
| Linear Steps | 10.4 | 3.1 | 16.3 | 3.8 | 13.4 | 4.3 | 28.5 | 6.5 | 13.4 | 4.0 | 28.5 | 6.5 |
| Auto-Reg [2] | 10.2 | 3.1 | 16.4 | 3.7 | 13.6 | 4.3 | 28.5 | 6.5 | 13.7 | 4.2 | 28.5 | 6.5 |
| Pruning Keysteps | 11.3 | 3.4 | 16.4 | 3.8 | 13.4 | 4.2 | 28.5 | 6.5 | 13.5 | 4.1 | 28.5 | 6.5 |
| Ours | **16.3** | **5.4** | **20.0** | **4.9** | **15.4** | **4.7** | **28.6** | **6.6** | **16.9** | **5.4** | **28.9** | **6.7** |
| | ± 0.3 | ± 0.1 | ± 0.2 | ± 0.1 | ± 0.1 | ± 0.1 | ± 0.0 | ± 0.0 | ± 0.1 | ± 0.1 | ± 0.1 | ± 0.0 |
| Ours w/ $\gamma = 0.3$ | 15.9 | 5.1 | 19.1 | 4.4 | 15.4 | 4.7 | 28.6 | 6.6 | - | - | - | - |
| Ours w/ $\gamma = 0.4$ | 16.1 | 5.2 | 19.4 | 4.7 | 15.1 | 4.6 | 28.6 | 6.6 | - | - | - | - |
| Ours w/ $\gamma = 0.5$ | **16.3** | **5.4** | **20.0** | **4.9** | 14.9 | 4.5 | 28.6 | 6.6 | - | - | - | - |
| Video-Text Fusion | - | - | - | - | - | - | - | - | 14.0 | 4.3 | 28.6 | 6.6 |
| BRF | 13.5 | 4.1 | 17.1 | 4.3 | 14.0 | 4.4 | **28.6** | **6.6** | 14.9 | 4.6 | 28.6 | 6.5 |

various thresholds compared to the baselines. We sweep the threshold in the range $[0.3, 0.5]$ at a step size of $0.05$.

## 5 Video-Text Keystep Recognition Ablations

We propose to use a priority-based keystep assignment where we use video signals in case both video and text modalities choose inconsistent keysteps (L245). We also experiment with video-text feature fusion. Table 1 shows the results. Fusion is worse than using video as priority because of inconsistent confident distribution across these modalities. Video features tend to show better performance at lower thresholds than text feature. We observe similar performance when using weighted fusion as well. In contrast, using video feature as priority solves this issue in a parameter-free way.

## 6 Keystep Recognition with Noisy Labels

We extend our setup to further demonstrate the robustness of our method in the presence of irrelevant keysteps. In this setup, we add keysteps from HowTo100M into the clean annotated keystep set of COIN and CrossTask to simulate an increasingly larger vocabulary. For each test dataset, we randomly select $\alpha N$ keysteps from the 10588 keysteps used in HowTo100M, where $\alpha$ is a scaling factor and $N$ is the number of keysteps in the test dataset, and inject these keysteps into test dataset vocabulary. This results in a much larger vocabulary size of $(1 + \alpha)N$. We progressively increase $\alpha$ and evaluate the performance at each scale level. We show in Table 2 the performance on text-only modality and we see the same trend in other modalities. The vocabulary size is shown in the first column (e.g., $1.5 \times N$ implies $\alpha = 0.5$). The frame-wise accuracy of the DistantSupervision [1] baseline and our method with the bigger vocabulary are shown in columns 2 and 3, respectively. Our relative improvement in accuracy over DistantSupervision [1] is shown in column 4.

## 7 Task Graph Ablations

Next we perform additional experiments to further accentuate the use of the task graph for zero-shot keystep recognition (Sec 4.1 of the main paper).

**Additional Ablations.** We propose a probabilistic video-mined graph $\mathcal{T}$ for keystep recognition. The graph is built using a constant threshold. We compare the performance if we alter each of these choices —using a non-probablistic graph and using an adaptive threshold instead of a fixed threshold. The adaptive threshold is per-video, i.e., the threshold is set such that 50% of the video is kept based on the similarity score and the remaining 50% are influenced by the graph transitions. Table 3 (a)

Table 2: Zero-shot keystep recognition under various noise levels for COIN and CrossTask dataset. Our method performs significantly better than the baseline even in the presence of strong noise.

(a) COIN dataset (N = 749)

| Vocab Size | DistantSup [1] | Ours | Relative Gain |
|---|---|---|---|
| 1 x N | 9.8 | 16.3 | 66% |
| 1.5 x N | 8.4 | 14.1 | 68% |
| 2 x N | 7.7 | 13.3 | 73% |
| 4 x N | 6.2 | 10.9 | 76% |
| 5 x N | 5.6 | 10.6 | 89% |
| 10 x N | 4.3 | 8.4 | 95% |

(b) CrossTask dataset (N = 105)

| Vocab Size | DistantSup [1] | Ours | Relative Gain |
|---|---|---|---|
| 1 x N | 16.1 | 20.1 | 25% |
| 1.5 x N | 13.3 | 16.8 | 26% |
| 2 x N | 12.6 | 15.7 | 25% |
| 4 x N | 10.1 | 12.9 | 27% |
| 5 x N | 8.9 | 11.7 | 31% |
| 10 x N | 7.5 | 10.1 | 35% |

shows the results. We observe that these ablations are inferior to our setting and we outperform both the baselines.

**Additional Metrics.** Table 3 (b) shows results for two additional metrics. Since our method results in a sequence of keystep predictions, edit distance (ED) is another suitable metric to compare our predicted sequence with the ground truth sequence. Similarly, F1 captures overall recall and precision performance, augmenting Accuracy and IoU already reported in Table 1 of the main paper. Our method outperforms all the baselines in the additional metrics as well.

**Keystep Recognition under High and Low Predictability.** To evaluate if our method works well even under low keystep predictability, given the set of low score predictions, we split it into two sets: one where the keystep predictability using the task graph prior is high, and one where it is low. To measure keystep predictability, we use the Shannon entropy of starting keystep. We choose a threshold such that the predictions are split into half. High Shannon entropy means low keystep predictability and vice-versa. Table 3 (c) shows the results. We see that our task graph outperforms the baselines even in the case of low predictability of keysteps. Of course, the gain is lower than it is for cases with high keystep predictability. Thus, our task graph prior can be seen as a way to correct noisy perceptual predictions, even in cases of low predictability.

# 8 Bayesian Recursive Filtering

We also evaluate the keystep assignment using Bayesian Recursive Filter (BRF) [3] instead of the proposed beam-search algorithm. The idea is that we model the similarity scores as a noisy measurements and then recursively update the predictions based on task graph transitions. Concretely, we maintain a $|\mathcal{K}|-$dimensional belief vector $\mathcal{B}(t)$ that denotes the probability distribution across all possible keysteps at time $t$. We compute similarity score $s(\hat{k}_t)$ same as our proposed algorithm (L202) and use the previous belief vector to update the current belief vector. We have, $k_t = \mathrm{argmax}\ \mathcal{B}(t)$ where $\mathcal{B}(t)$ is recursively calculated as

$$\mathcal{B}(t) = \begin{cases} s(\hat{k}_t) & \text{if } t = 0 \\ s(\hat{k}_t).A^T\mathcal{B}(t-1) + \epsilon s(\hat{k}_t) & \text{otherwise} \end{cases}$$

Here the multiplicative term $A^T\mathcal{B}(t-1)$ computes the likelihood of the current keystep from all previous keysteps. The last additive term is used to tune our confidence on belief vector vs similarity measurements.

Table 1 shows the result using this method. Clearly, this method has a lower performance than our proposed method. Nevertheless, the performance remains higher than the similarity-based baselined because of the meaninful updates in keystep assignment using BRF. Moreover, this method is causal, i.e. we only look backward in time to make future predictions. The performance remains low because this method uses lesser context and error made in earlier prediction rounds are propagated to all time instances after that.

Table 3: (a) Additional ablations, (b) additional metrics (ED and F1 score) and (c) splits of the performance for low and high entropy values.

| | Text-only | | | | Video-only | | | | Video-Text | | | |
| | COIN | | CrossTask | | COIN | | CrossTask | | COIN | | CrossTask | |
| **(a) Additional Ablations** | | | | | | | | | | | | |
| Method | Acc | IoU | Acc | IoU | Acc | IoU | Acc | IoU | Acc | IoU | Acc | IoU |
| Non-probabilistic | 10.5 | 3.2 | 16.4 | 3.8 | 13.6 | 4.2 | 28.5 | 6.5 | 13.7 | 4.1 | 28.5 | 6.5 |
| Adaptive Threshold | 12.5 | 3.8 | 17.1 | 4.0 | 14.1 | 4.3 | 28.5 | 6.5 | 14.0 | 4.4 | 28.6 | 6.5 |
| Ours | **16.3** | **5.4** | **20.0** | **4.9** | **15.4** | **4.7** | **28.6** | **6.6** | **16.9** | **5.4** | **28.9** | **6.7** |
| **(b) Normalized Edit Distance (ED) ↓ and F1 score** | | | | | | | | | | | | |
| Method | ED | F1 | ED | F1 | ED | F1 | ED | F1 | ED | F1 | ED | F1 |
| VideoCLIP [4] | - | - | - | - | 0.90 | 9.9 | 0.76 | 12.9 | 0.87 | 10.7 | 0.75 | 13.2 |
| DistantSup. [1] | 0.88 | 7.5 | 0.82 | 7.6 | - | - | - | - | - | - | - | - |
| Pruning Keysteps | 0.86 | 9.7 | 0.82 | 8.9 | 0.88 | 11.4 | 0.76 | 12.5 | 0.86 | 11.3 | 0.75 | 12.8 |
| Adaptive Threshold | 0.85 | 10.9 | 0.81 | 9.1 | 0.88 | 12.8 | 0.74 | 13.0 | 0.85 | 13.6 | 0.74 | 13.0 |
| Ours | **0.82** | **17.1** | **0.78** | **12.8** | **0.84** | **17.4** | **0.74** | **13.8** | **0.81** | **18.8** | **0.74** | **14.0** |
| **(c) Accuracy for splits of low and high Shannon entropy."Low" entropy means high predictability.** | | | | | | | | | | | | |
| Method | Low | High | Low | High | Low | High | Low | High | Low | High | Low | High |
| VideoCLIP [4] | - | - | - | - | 17.7 | 11.5 | 30.1 | 22.1 | 18.0 | 12.4 | 30.2 | 22.2 |
| DistantSup. [1] | 14.0 | 8.3 | 17.1 | 11.1 | - | - | - | - | - | - | - | - |
| Pruning Keysteps | 16.1 | 10.6 | 20.0 | 12.5 | 18.1 | 11.8 | 31.0 | 22.2 | 18.8 | 13.0 | 30.9 | 22.5 |
| Adaptive Threshold | 18.5 | 11.2 | 20.9 | 13.8 | 18.8 | 12.1 | 31.3 | 22.4 | 19.7 | 13.6 | 31.1 | 22.6 |
| Ours | **23.5** | **14.1** | **25.1** | **16.3** | **22.7** | **13.7** | **33.5** | **22.9** | **24.0** | **14.7** | **33.7** | **23.1** |

# 9 Compute Details

For zero-shot keystep recognition, we use one 32GB GPU since we use pretrained features from [1, 4]. Next, for representation learning, we use 128 GPUs for a total compute time of 12 hours. The experimental setting is same as [1]. Finally, for downstream training, we use 32 GPUs (each with 32 GB) for an average compute time of 9 hours across all experiments.

# 10 Limitations and Societal Impact

We propose a novel approach for keystep recognition using video-mined task graphs. Our method uses high-confidence anchors as basis for using the PathSearch algorithm. It is possible that the feature extractor $f(v_t, n_t)$ outputs a keystep with high confidence while being incorrect. For example, if the ASR narrations is *"Let me put side the toaster to create more space"* when demonstrating *"how to make tiramisu"* the high-confidence keystep can be *"run the toaster"* whereas the demonstration is unrelated to toaster. We claim that such noisy/unrelated signal is difficult to avoid in similarity based measures.

Video representation learning and keystep recognition in general could risk negative impacts if any bias from the dataset influences the representations. For example, COIN/CrossTask/HowTo100M are collected from YouTube and they may only contain videos having certain kinds of home and those with access to recording devices and microphones. Such biases could result in failures when these systems are deployed in a diverse set of environments. For example, keystep recognition in a cluttered and low-end kitchen might not work if the model is trained in clean and tidy kitchens. In addition, using these video representations for AR/VR applications may raise user privacy concerns, depending on how the dataset creators went about collecting the video samples.