# OpenReview forum: "Video-Mined Task Graphs for Keystep Recognition in Instructional Videos"
_NeurIPS.cc/2023/Conference — NeurIPS 2023 poster_

### Official Review · Reviewer_wv2a · 2023-07-03

**Soundness:** 2 fair
**Presentation:** 3 good
**Contribution:** 3 good
**Rating:** 6
**Confidence:** 4

**Summary:**

This paper aims to recognize the keysteps in instructional videos using automatically mined task graphs. These task graph is automatically discovered from a set of narrated instructional videos and contains all keysteps in a given vocabularly, i.e. it is not limited to a single task. This allows dependences between keysteps to be represented and helps move away from the requirement of prior work to have a strict ordering of keysteps in a task. The mined task graph is used for keystep recognition, task recognition and keystep forecasting on COIN and CrossTask.

**Strengths:**

- The approach of task graph mining eliminates need for prior keystep recognition work of having set linear order of keysteps.

***

- Benefits not constrained to task of keystep prediction, the paper also shows the video representation is useful for related but distinct tasks of key step forecasting and task classification

***

- Authors clearly understand task and nature of instructional videos well, I particularly liked the insights given on lines 42-45

***

- The method is well motivated

**Weaknesses:**

- The main weakness of the work is the lack of clarity in supervision used in relation to prior works, making it difficult to assess the suitability of the baselines used.
- Particularly as the proposed work to a vocabularly of keysteps as supervision in addition to the video's narrations. The predefined keystep vocabularly isn't clear until the beginning of the method section.
- The difference in supervision to prior works, e.g. [70], [82], [84], could be better explained in the related work. Currently not clear why these aren't explained and compared to without reading these papers in depth. Particularly [84] appears to only be supervised by the narration of the video which is also used by this paper in addition to the keystep vocabularly.
- I only looked at these three works as a sample, so it is very possible that there are other papers mentioned in related work with similar supervision levels to [82] and [84].
- From Table 1 of [84], [84] appears to outperform the proposed approach when comparing to the downstream step forecasting result of COIN in Table 4. This seems to also be the case for the results provided in Table 2 of [84].
- It also isn't clear if DistantSupervison [44] use the same keystep vocabularly as the proposed work. From figure 4 many of the mistakes made by [44] appear to be due to the name of the keystep, sometimes [44] even gives more information e.g. in press chest, as the keystep names appear to be less limited.
- Adding the supervision used to the results tables would greatly help assess this better.

***

- The importantance of the keystep vocabularly isn't evaluated
- My main question is whether the proposed method is robust to noise in the keystep vocabularly? I.e. how does the performance degrade with a larger keystep vocabularly containing keysteps which aren't used.
- It eems to work well for COIN and CrossTask with curated keystep vocab. For HowTo100M a presumably noisier vocab is used from WikiHow, however it would be much stronger if this effect was tested.

***

- Limited visualization of the task graph. The graph itself might be interesting and contain some insights for learning from instructional video as hinted on in lines 165-174. It would be useful to be able to see (a portion of) the mined task graph. A very small part is shown in Figure 3. However it should have been possible to visualize the full crosstask graph in supplementary.
- I recommend the authors include example task graph(s) in supplementary in future versions.

***

- From related work Paprika [85] sounds the most similar and it isn't clear why this work isn't compared to.
- It is a contemporaneous work, appeared online 31 March 2023, so this could be a valid reason. However, the footnote explains that [85] is using a different setting to [44].
- Comparing on the setting used by [85] would make the work stronger.

***

- Not factored into reviewing score as its a contempoaneous work but [A] has similarities to this paper so it might help to include a citation to [A] and explain the differences in a future version.

***

[A] StepFormer: Self-supervised Step Discovery and Localization in Instructional Videos. CVPR 2023.

**Questions:**

- How does the supervision used in this work compare with prior works? Particularly [44], [82] and [84].

- Why is [84] not compared to?

- How important is a clean and well-defined keystep vocabularly to the method performance?

The rebuttal responded to the majority of my concerns particularly, it improved clarity in the supervision used by the proposed and prior works,. visualization of the task graph, experiments on the expanded keystep vocabulary with relative gain over distant supervision particularly convincing. I hope these are included in the final version of the paper.

Since the rebuttal has addressed all my major concerns and I found no major concerns in the other reviews I have raised my rating to weak accept. While comparison to paprika isn't a reason to reject this work since it is concurrent, better explaining the differences or having some kind of numerical comparison would make this work stronger and help future readers.

**Limitations:**

A small section on limitations is present in the supplementary material. It can be improved by also considering negative societal impact.

---

> ### Author Rebuttal · Authors · 2023-08-09
>
> We thank the reviewer for their encouraging comments and insightful feedback.
>
> **Q1. Lack of clarity in supervision used vs. prior work**
>
> This seems to be a misunderstanding. All the baselines use the same level of supervision. Specifically, task graph construction does not use ground-truth keysteps in zero-shot keystep recognition (Sec 4.1). Similarly, in video representation learning (Sec 4.2), there is no annotation for keysteps hence the pseudo-labels are generated from similarity scores. In table 1, in addition to the keystep set, the supervision is narration (text-only), video feature (video-only) and both (video-text) for all baselines. Hence, all the comparisons are fair and baselines are a suitable comparison.
>
> **Q2. Use of a vocabulary of keystep in addition to narrations**
>
> The keystep vocabulary is the set of annotated keysteps in COIN/CrossTask. It is used by all the methods and not specific to our method. All the baselines and SOTA use the keystep vocabulary when assigning keysteps (L252-261). We have described keysteps (L20-27) in detail in the introduction. We appreciate the feedback, and we will accentuate this further in the text.
>
> **Q3. Difference in supervision to [70], [82], [84]**
>
> Note that [70, 84] are contemporaneous work per the conference policy (also see the global comment at the top). We will add a detailed comparison between the tasks and supervision used in these works. Here are the differences:
>
> [70, 82] - The task the authors solve is “procedure planning” which is distinct from keystep recognition and representation learning. The goal in procedure planning is given a start and end image, the model needs to generate a “plan” to achieve the final goal. This task is unrelated to ours and comparison with it is not possible.
>
> [84] - The paper proposes procedure-aware video representation. For supervision, they use videos and their narrations from HowTo100M. However, they use a text encoder trained with CLIP that uses 400 million (image, text) pairs. Hence, the level of supervision used is incomparable. Further, it is a contemporaneous work that appeared in CVPR 2023 *after* the NeurIPS deadline.
>
> **Q4. Empirical comparison to [84]**
>
> See above discussion about [84] – it is contemporaneous work published after we submitted our paper, and according to NeurIPS policy, a paper should not be rejected for not comparing to concurrent work. We will explore ways to relate the two methods empirically, though note that regardless of how they may each fare on video representation learning, our idea to enhance keystep recognition with the task graph remains novel with respect to [84].
>
> **Q5. Supervision used by DistantSupervision [44]**
>
> DistantSupervision [44] uses the same keystep vocabulary (L151). Since the keystep set is the same, the mistake cannot be due to the name of the keystep. Regarding figure 4, the keystep set between [44] and our work is the same and the yellow boxes (row A) represent ASR text of the narrations and not keysteps. Overall, our method uses exactly the same supervision as [44].
>
> **Q6. Add supervision used to results tables**
>
> All results in every table (1, 2, 3) use the same level of supervision, so we don’t add supervision explicitly in the tables. We will make sure to accentuate this in the paper.
>
> **Q7. Importance of keystep vocabulary and its robustness to noise**
>
> The proposed method uses a vocabulary set much larger than the set of keysteps for one task. Eg. an average task in COIN consists of 3.9 keysteps but the vocabulary set is 749 keysteps. Similarly, in HowTo100M representation learning (Sec 4.2), the vocabulary set is 10588 keysteps which is much larger than keystep set of any one task (and hence, quite noisy). Therefore, the method is robust to noise in keysteps and we indeed use keysteps that are not part of the task for a given video.
>
> **Q8. Visualization of the task graph**
>
> Thank you for the suggestion!  We add a portion of the mined task graph for CrossTask in the attached rebuttal PDF (Figure 1). It contains 61 out of 105 keysteps with top 5 transitions labeled (to avoid clutter). We do see some interesting properties (also see Fig 3 of main paper). For example, “lower jack” happens only after “brake on”, “close lid” after “open lid” whereas “cut cucumber” and “cut onion” can both happen after each other.
>
> **Q9. Comparison on the setting used by [85], a contemporaneous work, appeared online 31 March 2023**
>
> Thank you for noting that this work is contemporaneous. Regarding comparison with [85], apart from the fact that it is contemporaneous, we found that [85] and [44] use different experimental settings. We use the original setting of [44] for better reproducibility, given the detailed setup descriptions in that paper. In fact, DistantSup [44] performs much better in its original implementation compared to the adjusted settings reported in [85] (DistantSup [44] gets only 32.74 as reported in [85] vs the DistantSup authors’ originally reported 54.1 in [44] for COIN step classification, similarly 82.66 vs 90.0 for COIN task classification; see DS* in Table 1 of [85] vs results in [44]). Hence, there is some disadvantage to DistantSup [44] in the revised setting deployed in [85], though the reason is not elaborated in [85]’s paper. Therefore, we stick to comparison with [44] (where the existing method DistantSup achieves its better numbers) for fairness and better reproducibility.  We observe clear gains over those numbers (Table 4 in the main paper).
>
> **Q10. [A] is a similar work**
>
> Thanks for pointing out concurrent [A]. It uses non-annotated data for discovering steps in instructional videos and shows downstream keystep localization (different than our task). We will add it in the final version of the paper and discuss the differences.
>
> **Q11. Societal impact**
>
> Please see the global comment above. Though we gave it careful thought, we do not see any negative societal impact specific to our contributions in this paper.

---

> > ### Comment · Reviewer_wv2a · 2023-08-17
> > **Response to rebuttal**
> >
> > Thank you for highlighting the improved clarity in the supervision used by the proposed and prior works. I hope this clarity can also be added to a future version of the paper. I also appreciated the visualization of the task graph in the rebuttal.
> >
> > It is a shame the authors could not experiment with the keystep vocabulary. I understand the vocabulary used within a task is much smaller than the vocabulary available to the model, however the available vocabulary, in COIN for instance, is still much smaller than the one used in HowTo100M and much smaller than the vocabulary available in the English language. It is also a shame the authors refuse to consider any potential negative societal impact.
> >
> > Nonetheless, I will enter the discussion phase more positive than my initial review thanks to the clarity on supervision.

---

> > > ### Author Response · Authors · 2023-08-19
> > > **Thanks and follow-ups**
> > >
> > > We thank the reviewer for acknowledging our explanation of the supervision and visualization of the task graph. We are happy that the reviewer is **more positive** about the paper now. We answer the two questions raised by the reviewer below:
> > >
> > > **Q1. Experiment with the keystep vocabulary**
> > >
> > > We thank the reviewer for observing that we use a much larger vocabulary than that used within a task in COIN/CrossTask. This choice of larger keystep set shows the robustness of our method in the presence of irrelevant keysteps as we discussed above.
> > >
> > > To further expand on the specific point of the reviewer *“how does the performance degrade with a larger keystep vocabulary containing keysteps which aren't used.”* – we extend our setup to further demonstrate the robustness of our method in the presence of irrelevant keysteps. In this setup, we add keysteps from HowTo100M into the clean annotated keystep set of COIN and CrossTask to simulate an increasingly larger vocabulary. For each test dataset, we randomly select $\alpha N$ keysteps from the 10588 keysteps used in HowTo100M, where $\alpha$ is a scaling factor and $N$ is the number of keysteps in the test dataset, and inject these keysteps into test dataset vocabulary. This results in a much larger vocabulary size of $(1 + \alpha) \times N$. We progressively increase $\alpha$ and evaluate the performance at each scale level. We show here the performance on text-only modality and we see the same trend in other modalities. The vocabulary size is shown in the first column (e.g., 1.5 x N implies $\alpha = 0.5$). The frame-wise accuracy of the DistantSupervision baseline and our method with the bigger vocabulary are shown in columns 2 and 3, respectively. Our relative improvement in accuracy over DistantSupervision [44] is shown in column 4.
> > >
> > >
> > > **Zero-shot keystep recognition on COIN dataset (N = 749)**
> > > | Vocabulary size      | DistantSupervision [44] | Ours | Relative Gain |
> > > | ----------- | :-----------: | :------------: | :-----------: |
> > > | 1 x N (original set)      |  9.8  | 16.3 | 66% |
> > > | 1.5 x N  |  8.4 | 14.1 | 68% |
> > > | 2 x N  |  7.7 | 13.3 | 73% |
> > > | 4 x N  |  6.2 | 10.9 | 76% |
> > > | 5 x N  |  5.6 | 10.6 | 89% |
> > > | 10 x N  |  4.3 | 8.4 | 95% |
> > >
> > > _______________________________________
> > >
> > > **Zero-shot keystep recognition on CrossTask dataset (N = 105)**
> > > | Vocabulary size      | DistantSupervision [44] | Ours | Relative Gain |
> > > | ----------- | :-----------: | :------------: | :-----------: |
> > > | 1 x N (original set)      |  16.1  | 20.1 | 25% |
> > > | 1.5 x N  |  13.3 | 16.8 | 26% |
> > > | 2 x N  |  12.6 | 15.7 | 25% |
> > > | 4 x N  |  10.1 | 12.9 | 27% |
> > > | 5 x N  |  8.9 | 11.7 | 31% |
> > > | 10 x N  |  7.5 | 10.1 | 35% |
> > >
> > > _______________________
> > >
> > > As expected, having a bigger keystep vocabulary containing irrelevant keysteps deteriorates the performance of both methods (see columns 2 and 3). The more irrelevant keysteps, the larger the reduction in performance. Nevertheless, our method is noticeably more robust to large keystep sets when compared to the baseline, and its advantage over the baseline steadily increases with the increasing number of keysteps (see column 4).
> > >
> > > This demonstrates the regularization power of our task graph, which corrects the noisy similarity-based predictions with task graph priors (L196-200 in the main paper), thus resulting in a robust keystep recognition. These results empirically demonstrate the advantage of our task graphs under a much larger vocabulary setting. We will elaborate this result in the final version of the paper.
> > >
> > >
> > > **Q2. Societal Impact**
> > >
> > > We appreciate the reviewer’s suggestion to delve deeper here.  Video representation learning and keystep recognition in general could risk negative impacts if any bias from the dataset influences the representations. For example, COIN/CrossTask/HowTo100M are collected from YouTube and they may only contain videos having certain kinds of home and those with access to recording devices and microphones. Such biases could result in failures when these systems are deployed in a diverse set of environments. For example, keystep recognition in a cluttered and low-end kitchen might not work if the model is trained in clean and tidy kitchens. In addition, using these video representations for AR/VR applications may raise user privacy concerns, depending on how the dataset creators went about collecting the video samples. We will emphasize these in our final draft.

---

> > > > ### Comment · Reviewer_wv2a · 2023-08-22
> > > > **Response to follow-ups**
> > > >
> > > > Thanks for responding to my remaining concerns. I find the experiments on the expanded keystep vocabulary with relative gain over distant supervision particularly convincing. I hope these are included in the final version of the paper.
> > > >
> > > > Since the rebuttal has addressed all my major concerns I will raise my rating to weak accept.

---

### Official Review · Reviewer_qtTp · 2023-07-04

**Soundness:** 3 good
**Presentation:** 3 good
**Contribution:** 3 good
**Rating:** 5
**Confidence:** 4

**Summary:**

In this work, the authors propose to address keystep recognition in instructional videos. To achieve this goal, they attempt to build a task graph from videos, which show how keysteps are related to each other. Based on this graph, one can further update the preliminary keystep assignment, when the initial prediction is with low confidence.


**Strengths:**

1 The key step recongnition is an important topic for procedural activity understanding.
2 The idea of building task graph seems to be technical sound as key step relation prior.
3 The experiments show the effectiveness of the method.

**Weaknesses:**

(1) The task graph is pre-computed offline or built online? I assume, it should be built offline, according to the unannotated dataset of narrated instructional videos. Moreover, when working on another dataset, the task graph should be re-computed again?

(2) What is the computation time of Path Search, when updating the low-confident key step prediction?

**Questions:**

Please see the weakness section.

**Limitations:**

Please see the weakness section.

---

> ### Author Rebuttal · Authors · 2023-08-09
>
> We thank the reviewer for their encouraging comments and insightful feedback.
>
> **Q1. The task graph is pre-computed offline or built online? I assume, it should be built offline, according to the unannotated dataset of narrated instructional videos. Moreover, when working on another dataset, the task graph should be re-computed again?**
>
> Yes, the task graph is pre-computed offline. When working on a dataset with a different vocabulary, the task graph needs to be re-computed to accommodate for the difference in ground truth keystep set. We generalize with HowTo100M where we use a large-scale keystep set (>10k keysteps). More precisely, if the keystep set were the same in all dataset, we would not have to compute the task graph again.
>
> **Q2. What is the computation time of Path Search, when updating the low-confident key step prediction?**
>
>
> PathSearch uses Dijkstra’s algorithm (L208) that has a worst-case complexity of $O(V^2)$ for $V$ nodes (equal to number of keysteps, $V = 105/749/10588$ for CrossTask/COIN/HowTo100M). This computation is minimal. When performing keystep assignment, PathSearch is less compute intensive than video feature extraction since forward pass is typically more than just $O(V^2)$ hence no delay is observed compared to the baselines.  Hence our idea adds minimal overhead while achieving notable advantages in keystep recognition and video representation pretraining.

---

### Official Review · Reviewer_vrnW · 2023-07-06

**Soundness:** 2 fair
**Presentation:** 3 good
**Contribution:** 2 fair
**Rating:** 4
**Confidence:** 5

**Summary:**

This paper addresses the problem of key-step recognition and localization in instructional videos by learning and leveraging a probabilistic task graph. The proposed method first localizes key-steps mined from text sources (such as wikihow) in videos by measuring the similarity between visual-narration features in videos (obtained from pretrained models) and key-step feature. It then constructs a graph whose nodes are key-steps and whose edges are transitions between key-steps (obtained from the localization results). Finally, the initial key-steps whose confidence is lower than a threshold will be replaced by the key-steps from the optimal key-step path in the task graph. The experimental results show some improvement compared to key-step recognition baselines.

* The reviewer read the author rebuttal and other reviews.

**Strengths:**

- The paper is easy to read and overall framework is sufficiently well presented.

- Leveraging a task graph to improve recognition is interesting (although the paper is not the first work addressing it).

**Weaknesses:**

-  The paper claims that it is the first to use task graphs to enhance keystep predictions in instructional video (see line50). The reviewer disagrees with this claim. Parika [85] has addressed learning and leveraging task graphs to learn video representation for better key-step recognition. The final goal that the submission and Parika pursue is almost the same. In fact the submission has a narrower scope compared to [85] as it does not address representation learning while [85] addresses it along with better key-step recognition.

- The graph learned in the paper is not exactly a "task graph" (a task graph encodes all possible ways of doing a task). It is rather a transition probability model between key-steps. In particular, a limitation of the transition model in the paper is that it does not properly encode task executions, e.g., a transition model allows multiple transitions between two key-steps (e.g., a-->b-->a-->...-->b-->a), which may be invalid for task execution. This is because of the short-sightedness of the transition model that does not model long-range action dependencies.

- Related to the comment above, there is a need for an experiment that measures the edit distance between the predicted key-step sequences and the ground-truth key-step sequences, compared with baselines.

- Compared to [85] which also builds a task graph, the advantage of the studied method is not clear. Given the high similarity between the two works, there is a need to include [85] with its experimental setting in the paper for a fair comparison.

- The key-step to narration assignment can lead to violation of the transition model (see the formulation after line 205). Specifically, when the similarity between video and key-step is above a threshold, the key-step will be assigned to the video frame and is lower than a threshold it will be assigned by following the transition model. Additionally, there is a need in the experiment that shows the effect of the hyper-parameter $\gamma$ on the step recognition results (e.g., horizontal axis $\gamma$ and vertical axis $acc$).

- The evaluation metric in the paper does not consider background. Given that at least 40% of instructional videos in crosstalk and coin consist of background frames, there is a need to measure how well the proposed method avoids assigning key-step labels to the background frames. By measuring ACC and IOU for only key-step regions, one cannot evaluate whether the paper wrt SOTA assigns large or small portions of background frames as key-steps.

- In Table 1, for Video-only and Video-Text, both ACC and IOU of the method is very close to SOTA, which makes the effectiveness of the learn transition model questionable.

**Questions:**

In addition to the questions raised in the weakness section:

- From Table 1, one can se that the improvement of the performance on COIN is often  more significant than on CrossTask. Why?

- Is the task graph built based on the videos and narrations in the training set and uses the same task graph during test? Or is the task graph only built based on the test set?

**Limitations:**

There is no discussion of limitations and possible negative societal aspects of the work and need to be included in the main paper.

---

> ### Author Rebuttal · Authors · 2023-08-10
>
> Thanks for the valuable comments. Our responses show where multiple requests from the reviewer were addressed in the submitted paper.  Also please note the concurrent work policy of NeurIPS re: [85].
>
> **Q1. Comparison with [85]**
>
> Paprika [85] is a contemporaneous work per the conference guidelines (see global rebuttal), meaning a comparison is not required.
>
> Regarding relative contributions: As we were completing our submission, we saw their paper appear on arXiv; we explain the important differences with [85] in L96-105. Paprika learns a task graph ONLY for video representation learning, whereas we use the task graph for both keystep recognition (Sec 4.1) and video representation learning (Sec 4.2). Paprika does not perform zero-shot keystep recognition. Our work is the first to use task graphs to enhance keystep predictions. Hence our scope is broader than [85].
>
> Regarding empirical comparisons: In [85], the resulting task graph has equi-probable edges, whereas ours is based on empirically observed transitions in video. A non-probabilistic baseline in Table 1(a) in the attached rebuttal PDF shows that a probabilistic graph performs better.  Important: please see our response to Q9 for Reviewer wv2a discussing issues with direct comparison with [85] on representation learning.
>
> **Q2. Transition probabilities vs “task graph” e.g. transition model allows multiple transitions between two key-steps, which may be invalid**
>
> As a counterexample to the reviewer’s claim, keystep a: “add salt to boiling soup” and keystep b: “stir the soup to dissolve salt” can have a→b→a→b…. transitions when a user is not sure about how much salt to add and hence this transition cannot be disallowed. Specifically, given the stochastic nature of human activity, it is impossible to encode ALL possible ways of doing a task. Thus, a transition probability model is a reasonable way to capture all typical transitions, also helping accommodate transitions outside a fixed recipe. Note that the PathSearch algorithm (L205-211) finds a keystep path between high confidence keysteps, regardless of the duration in between, i.e., our method can also consider long-range action dependencies.
>
> Please see the “Linear Script” baseline (L289-290) where we take the explicit order of steps. Also see the “Auto-Regressive” (L280-282) baseline that has long-range action dependencies and still performs worse. Both these baselines show that our explicit task graph is helpful in assigning keysteps.
>
> **Q3. Add edit distance as a metric**
>
> Thanks for the suggestion. We think edit distance is a useful additional metric, since it captures how similar the sequence of keysteps are w.r.t. the ground truth. Please see Table 1(b)  in the attached PDF. On this metric, too, our method outperforms baselines for all modalities for both datasets. We will add this metric in the paper.
>
> **Q4. Violation of transition model (L205) and effect of $\gamma$**
>
> Please note that Supp. Table 1 already includes the suggested experiment with $\gamma$. See also L245.
>
> Regarding transition violations – the algorithm first chooses instances where the similarity between video and key-step is above a threshold and between any two such time instants, we use the transition model (L203-205). We also have another version of the transition model - Bayesian Recursive Filter (BRF), (please see footnote 2 of the main paper and section 6 of the supp). In this version, we make predictions based only on the transition model and we observe that the performance in that case is weaker. Applying the transition model only between two high confidence segments is empirically better. Establishing this was part of our research exploration, since we originally pursued both models in parallel.
>
> **Q5. Metrics do not consider background**
>
> We also have metrics that do consider background. The chosen setting is consistent with prior work [19]. Accuracy metric is not ideal using background due to class imbalance (simply predicting everything as background results in high Acc). Note that IoU (reported in our submission) does consider background and penalizes false positives. Moreover, we add F1 metrics to further address the background frames question. See Table 1(b) in the attached rebuttal PDF. Our method outperforms all the baselines on all metrics consistently, including IoU and F1, which consider background.
>
> **Q6. Performance close to SOTA in Video-only and Video-Text**
>
> The increase in Acc for COIN dataset is 2.2% and 3.6% for video-only and video-text wrt SOTA. The difference is indeed lower for CrossTask while still being statistically significant. We attribute this to the smaller keystep set in CrossTask (105 vs 749 keysteps in COIN) and thus the transitions having lower predictability on average. The video features may themselves have low perceptual errors as we still see a gain of 3.9% in CrossTask text-only. As suggested by reviewer G41M, if we split the performance into low and high predictability segments, we see that the performance gain in highly predictable keystep transitions is more significant (3.4% for video-only and 3.5% for video-text). Therefore, our method performs better with a larger keystep set — important for the natural setting.
>
> **Q7. Gain on COIN more than CrossTask**
>
> Please see response to Q6. In short, COIN offers a much larger keystep taxonomy, and hence a greater need for our regularization.
>
> **Q8. Construction of task graph on training or test set?**
>
> For zero-shot keystep recognition (Sec 4.1), the task graph is built only based on the test set since we do not want supervision transfer between different splits. For representation learning (Sec 4.2), the task graph is built on the train set to pseudo label itself.
>
> **Q9. Limitations**
>
> We have discussed limitations in Supp (L59-65). Also see the global rebuttal. Though we've given it careful thought, we do not see negative societal impact for our contributions.

---

> ### Comment · Reviewer_vrnW · 2023-08-22
>
> Thanks for your responses above. I believe the rebuttal addressed several important comments I raised in my review. Additionally, the point about concurrent work [85] is noted/valid. I will enter the discussion phase being more positive about this submission and willing to increase my score.

---

### Official Review · Reviewer_G41M · 2023-07-08

**Soundness:** 3 good
**Presentation:** 3 good
**Contribution:** 3 good
**Rating:** 6
**Confidence:** 3

**Summary:**

The paper considers the task of "keystep" recognition. Keystep is one of the N sub-tasks that are performed sequentially to achieve a goal / task. Keysteps can have causal dependencies. Prior work has the following limitations:
(1) only considers each keystep in isolation, without considering the overall task (sequence of keysteps) [this is suboptimal, especially when powerful models like Transformers can efficiently model temporal context]
(2) Keysteps are expected to conform exactly to a predetermined script of actions [sequence is not always precisely defined — there can be order inversions, skips, alternate keysteps, etc.]
(3) Classification of fixed-size, pre-segmented chunks [this is inaccurate].

The paper presents an approach to probabilistically represent the keystep transitions for tasks in the form of a "task graph". The approach leverages the prior probabilities for keystep transitions in the task graph in cases where there is low-confidence for video evidence. This results in improvements in performance in the zero-shot keystep localization task. It further improves video representations for other downstream tasks.

**Strengths:**

(1) The proposed approach explicitly leverages the probabilities of the task graph. This has the advantage that it is more interpretable than, e.g., encoding the relevant nodes and keysteps as an embedding, and then making predictions based on this.

(2) There is consistent performance improvements in all tasks, especially zero-shot keystep prediction. Also, a big plus is the improvement in downstream tasks that leverage the better video representation that is trained on predicted pseudo-labels for other tasks.

(3) It is nice that the task graph is constructed across the entire dataset, and contains common keysteps across tasks. This makes the task graph extensible across (theoretically) infinite data. With a much larger task graph, one could encode the keysteps for a majority of human activities present in videos.

**Weaknesses:**

(1) The approach involves constructing a task graph that probabilistically models the transitions between keysteps. Inference on this task-graph is also done by a principled path-search algorithm. However, to incorporate the probabilities from the task graph, a simple confidence-based thresholding operation is proposed. I.e., if the evidence (confident predictions) are below a certain threshold, then fall-back to the prior. This seems simplistic, and might not capture the complexity of individual data samples. Do we expect that a single threshold value across an entire dataset is a reasonable choice? Alternately, is there a way to have continuous bayesian approach incorporating the evidence and the prior for all samples?

(2) Solely relying on priors for low-confidence perceptual predictions can work for tasks with very high predictability of keysteps. For almost all others, this might cause large errors. Is there a breakdown of errors from the perceptual model and the task-graph-prior model? Please include this in future versions of the paper.

(3) Keystep classification on a temporal span of exactly 1-second-long clips is arbitrary. What’s the distribution of keystep temporal spans? Prior work indeed does this, but it would be nice to progress beyond this. However, I recognize that this may become very challenging in the case of novel or rare keysteps.

**Questions:**

(1) Frame-wise accuracy metric is defined (in L264) as “the fraction of frames with ground truth k_i that has the correct assignment.” Accuracy metrics usually penalize both false negatives and false positives. The definition of frame-wise accuracy suggests that this is a recall-like metric (only penalizes false negatives). Am I interpreting the metric definition correctly?

(2) A question that arises wrt Strengths (1) is — is using the task graph probabilities explicitly, better than providing the relevant nodes in the task graph as input condition / context to a “revise” model (similar to baseline “Auto-Regressive [65, 69]”)?

**Limitations:**

Not explicitly discussed in the main paper.

---

> ### Author Rebuttal · Authors · 2023-08-09
>
> We thank the reviewer for encouraging comments and insightful feedback.
>
> **Q1. Do we expect that a single threshold value across an entire dataset is a reasonable choice? Alternately, is there a way to have continuous bayesian approach incorporating the evidence and the prior for all samples?**
>
> Thanks for the question.  Our submission did consider this.  We provided experiments with a Bayesian Recursive Filter (BRF) variant of our approach that does consider the evidence and prior for all samples.  See section 6 and Table 1 in supplementary and Footnote 2. However, the performance is lower than the proposed method because BRF is causal and hence observes less context, and errors made in earlier prediction rounds are propagated to all time instances after that (Supp L49-52). The proposed simpler method is empirically better than the more complicated BRF. Establishing this was part of our research exploration, as we originally pursued both models in parallel.
>
> As discussed in L245, performance is not highly sensitive to choices of the threshold $\gamma$ in the range [0.3, 0.5] (also see ablation in Supp. Table 1).  We experimented with adaptive thresholding and found the performance to be inferior to using a constant threshold. The result can be seen in Table 1(a) in the attached rebuttal PDF. We adaptively identify one threshold per video, such that 50% of the video clips are considered to have “high-confidence” similarity scores. This works even for videos dominated by low similarity scores. The performance drops in this setting. We attribute this decrease to the fact that the model is assigning noisy samples as the high-confidence keystep and hence the overall keystep prediction is affected. We will add this ablation in the final paper.
>
> **Q2. Solely relying on priors for low-confidence perceptual predictions can work for tasks with very high predictability of keysteps. For almost all others, this might cause large errors. Is there a breakdown of errors from the perceptual model and the task-graph-prior model?**
>
> Thanks for suggesting this breakdown of errors. First, given the set of low-confidence perceptual predictions, we split it into two sets – first where the keystep predictability using task graph prior is high and the second where it is low. To measure keystep predictability, we use Shannon entropy of starting keystep. We choose a threshold such that the predictions are split into half. High Shannon entropy means low keystep predictability and vice-versa. Table 1(c) in the attached rebuttal PDF shows the performance. We see that our task graph outperforms the baselines even in the case of low predictability of keysteps. Of course, the gain is lower than it is for cases with high keystep predictability. Thus, our task graph prior can be seen as a way to correct noisy perceptual predictions, even in cases of low predictability. We will add this observation in our final paper.
>
> **Q3. Keystep classification on a temporal span of exactly 1-second-long clips is arbitrary. What’s the distribution of keystep temporal spans? Prior work indeed does this, but it would be nice to progress beyond this. However, I recognize that this may become very challenging in the case of novel or rare keysteps**
>
> Average temporal span of a keystep is 14.91s and 9.61s for COIN and CrossTask, respectively. Our setting is consistent with prior work [78,47]. Choosing non-overlapping contiguous segments of 1 seconds makes the keystep classification uniform and independent of keystep duration (some are long and some are short). For longer keysteps, the model can classify the same keystep for contiguous segments, denoting a longer temporal span of a particular keystep. So, in short, processing 1-second clips does not preclude identifying longer keystep instances when they occur. Work progressing beyond 1s video clips is orthogonal to our contributions.
>
> **Q4. Accuracy metrics usually penalize both false negatives and false positives. The definition of frame-wise accuracy suggests that this is a recall-like metric (only penalizes false negatives). Am I interpreting the metric definition correctly?**
>
> The definition in L264 only means that the ground truth set does not contain non-labelled (i.e. background) segments. The accuracy is the standard (Correct predictions)/(All non-background segments). Concretely, if $\phi$ is the background label, then we use the standard multi-class accuracy as
>
> $Acc = \frac{\sum \limits_{V \in D} \sum \limits_{t \in |V|} 1(\hat{k}_t = k_t)}{ \sum 1(k_t \neq \phi)}$
>
> Here, D is the dataset and V is video. This setting is consistent with prior work [19]. Using background in the accuracy metric is not ideal due to class imbalance (>50% background in both COIN and CrossTask). Note that IoU (another metric we report in Table 1) still penalizes false positives that are in the background. In the denominator of IoU we do add the false positives. We also add the F1 metric (including the background) for completeness in the rebuttal PDF Table 1(b). Our method outperforms the baselines and SOTA in all metrics.
>
> **Q5. A question that arises wrt Strengths (1) is — is using the task graph probabilities explicitly, better than providing the relevant nodes in the task graph as input condition / context to a “revise” model (similar to baseline “Auto-Regressive [65, 69]”)?**
>
> Please note that we provided a baseline that addresses this question.  Pruning keysteps (L283-288) is the baseline that first clusters the keysteps and tries to find the “relevant nodes” and provides only those to the “revise” model. The performance is stronger than SOTA and auto-regressive, but still weaker than our method. Thus, explicit task graph probabilities are better than both auto-regressive baseline and providing only “relevant nodes”.
>
> **Q6. Limitation not explicitly discussed in the main paper.**
>
> We have discussed limitations in Supp (L59-65). Please also see the global comment above.

---

### Official Review · Reviewer_iD7r · 2023-07-10

**Soundness:** 3 good
**Presentation:** 3 good
**Contribution:** 3 good
**Rating:** 5
**Confidence:** 3

**Summary:**

This paper focuses on the procedural activity understanding task. Considering that the feature-keystep matching in current methods is independent and fails to encapsulate the rich variety, the authors propose a video-mined task graph as a prior to update the preliminary keystep assignments.

**Strengths:**

1. Using the graph structure to generate broader context and assign the correct keystep labels is an intuitive idea.

2. The propsed method offers substantial gains over state-of-the-art methods.

**Weaknesses:**

1.  What is the cost for building the video-mined graph? Will this graph leads to great overhead?

2. The authors use the graph structure to amplify the context receptive field. How about using the attention mechanism (e.g. self-attention) since it is also a relation extraction manner?

**Questions:**

1.  What is the cost for building the video-mined graph? Will this graph leads to great overhead?

2. The authors use the graph structure to amplify the context receptive field. How about using the attention mechanism (e.g. self-attention) since it is also a relation extraction manner?

**Limitations:**

1.  What is the cost for building the video-mined graph? Will this graph leads to great overhead?

2. The authors use the graph structure to amplify the context receptive field. How about using the attention mechanism (e.g. self-attention) since it is also a relation extraction manner?

---

> ### Author Rebuttal · Authors · 2023-08-09
>
> We thank the reviewer for encouraging comments and feedback.
>
> **Q1. What is the cost for building the video-mined graph? Will this graph leads to great overhead?**
>
> The overhead is minimal since the only extra computation is counting transitions given similarity scores followed by averaging. There are only $Nd$ additional operations for N-videos, each d-seconds long followed by averaging across $\mathcal{K}$ keystep classes. In fact, the cost of building the graph is comparable to a forward pass on only one test video that involves significantly more FLOPs. Hence our idea adds minimal overhead while achieving notable advantages in both keystep recognition and video representation pretraining.
>
> **Q2. The authors use the graph structure to amplify the context receptive field. How about using the attention mechanism (e.g. self-attention) since it is also a relation extraction manner?**
>
> The auto-regressive baseline (L280 and Table 1) does exactly this self-attention. In this baseline, an auto-regressive model first learns the transitions of the predicted keysteps. In the refining step, the model takes previous predictions and the current keystep-video score and generates the next prediction, typical of a self-attention mechanism. Our explicit representation significantly outperforms this baseline. It can be concluded that the proposed explicit task graph prior is more effective than self-attention, despite its implicit context modeling.

---

### Author Rebuttal · Authors · 2023-08-09

We thank all the reviewers for their insightful comments.

**Three reviewers lean towards acceptance**. We believe our responses below clarify the items raised in the reviews, which include questions about metrics and a concurrent paper [85] (*vrnW*) and a possible misunderstanding about supervision (*wv2a*). There are several instances where requested experiments/baselines are indeed already in our original submission, detailed below.

All the reviewers are positive about the method – *iD7r* states the method is intuitive, *G41M* calls it more interpretable, *vrnW* suggests the method is technically sound and finally, *wv2a* agrees that it is well-motivated. Further, reviewer *iD7r* states that “proposed method offers substantial gains over state-of-the-art methods” and *qtTp* says the experiments “show effectiveness”. *G41M* agrees that “there is consistent performance improvements in all tasks, especially zero-shot keystep prediction” and that improvement in downstream tasks is a “big plus” by using a “better video representation”. *G41M* correctly notes that the task graph is constructed across the entire dataset and “with a much larger task graph, one could encode the key steps for a majority of human activities present in videos.” *wv2a* notes that the “benefits are not constrained to task of keystep prediction but distinct tasks of key step forecasting and task classification”. Finally, we thank *vrnW* for noting that the paper is easy to read and to *wv2a* to note that we “clearly understand task and nature of instructional videos well”.

We address the weakness individually to each reviewer alongside their review.

**We would also like to draw attention of the AC and the reviewers to several contemporaneous works [70, 84, 85, A] (thanks to *wv2a* for pointing out [A])**. All of them are published in CVPR 2023 (**after** this paper deadline in June 2023) and appeared first online (on arXiv) within two months of the paper submission (i.e. after March 17th 2023). This means they are concurrent work per the NeurIPS policy given in the call for papers:.  **“...for the purpose of the reviewing process, papers that appeared online within two months of a submission will generally be considered "contemporaneous" in the sense that the submission will not be rejected on the basis of the comparison to contemporaneous work.”**

That said, all of them use different ideas for video representation learning. [85] is most relevant to our work, and (upon seeing their arxiv post as we were finishing our submission) we have differentiated with it in the submitted paper in detail (see L96-L105). These contemporaneous works further suggest the importance of video representation learning and using task graphs for keystep localization.


Finally, we have included a **Limitations section in the supplementary (Supp L59-65)**. As far as dependencies, our method relies on the strength of the perceptual prediction model, and our task graph construction and keystep prediction can be noisy if the perceptual model is weak. We do not foresee any negative societal impact.

---

### Decision · Program_Chairs · 2023-09-21

**Decision:**

Accept (poster)

**Comment:**

This method proposes an approach to compute transition probabilities between keysteps for instructional videos. Reviewers appreciated that the method was well motivated, and proposes an interesting formulation to the problem of keystep recognization and localization. There are some concerns about the actual inference on the task graph, which thresholds confidences, but the authors argue that they tried a more probabilistic approach but found that the results were worse. One reviewer was also concerned about whether the technical formulation of the task-graph is correct but after some discussion the ACs agree that being able to repeatedly transition to the same step is a reasonable assumption.